# REFLECTION ON KNOWLEDGE GRAPH FOR LARGE LANGUAGE MODELS REASONING

## ABSTRACT

Recent studies have highlighted the potential benefits of supplementing Large Language Models (LLMs) with information retrieved from knowledge graphs to enhance their performance. However, current approaches often introduce additional noise in the pipeline process of knowledge retrieval and reasoning, leading to the accumulation of errors, impeding LLMs from effectively combining the external knowledge in answering complex multi-hop questions. To this end, we introduce RefKG, an innovative framework specifically crafted to enhance the reasoning capabilities of LLMs through reflective engagement with knowledge graphs. In particular, RefKG autonomously conduct retrieval and reflection on knowledge graphs. Its reasoning process includes four steps: decomposing complex queries, retrieving and pruning evidence subgraphs, generating textual evidence, and evidence-enhanced reasoning. To enhance the alignment of LLMs with external knowledge, we have developed a multi-task tuning strategy that not only infuses knowledge to LLMs but also teaches them how to utilize the knowledge in answering questions, thereby significantly improving their ability to handle knowledge-intensive tasks. Experimental results on fact verification and knowledge graph question answering tasks demonstrate that RefKG outperforms previous state-of-the-art models.

## 1 INTRODUCTION

Large Language Models (LLMs) have recently showcased their robust and wide-ranging capabilities in tackling diverse challenges in NLP including machine translation (Zhang et al., 2023) and information extraction (Sainz et al., 2023). However, given the ever-evolving nature of real-world knowledge (Zhang et al., 2023), LLMs occasionally exhibit limitations in domain-specific expertise or in timely updating their knowledge bases. This shortfall often results in hallucinations within their responses, where the generated content deviates from factual accuracy (Huang et al., 2023).

To alleviate the issue of hallucinations in LLMs on knowledge-intensive tasks such as Knowledge Graph Question Answering (KGQA) (Gupta et al., 2018), a promising strategy involves augmenting LLMs with external knowledge sources (Tan et al., 2023), like knowledge graphs (KGs) (Luo et al., 2018; Hu et al., 2018). This augmentation process typically involves retrieving factually relevant knowledge from extensive knowledge bases to assist LLMs in formulating answers that rely on this externally retrieved information. However, existing solutions still suffers from several shortcomings. First, due to the large scale and complexity of knowledge graphs, the retrieval and reasoning processes often introduce numerous noise triplets that are irrelevant to the query. Prior approaches have been limited in their capacity to accurately address complex queries (Lan et al., 2021) and have struggled to effectively address such noise. Effectively, stably, and interpretably constraining retrieved subgraphs continues to be a critical challenge. Second, recent investigations (Li et al., 2023b; Nie et al., 2023) have predominantly performed black-box testing on proprietary models such as ChatGPT. These studies often employ in-context learning techniques (Liu et al., 2022), where external knowledge is incorporated into the prompts to steer the model's response generation. Although these training-free methods enable the integration of external knowledge, they do not enhance the interactive capabilities between LLMs and knowledge graphs, thereby limiting the potential of LLMs to efficiently acquire and deploy knowledge, especially when supervised signals are available. Additionally, black-box models cannot be deployed privately, which significantly limits their flexibility and adaptability.

Table 1: Comparison of different KGQA methods.

| Method | Query Decoupling | Knowledge Refinment | Knowledge Reconstruction | Multi-task Tuning |
|---|:---:|:---:|:---:|:---:|
| Retrieve-Rewrite(Wu et al., 2023)[IJCKG23] | ✗ | ✗ | ✓ | ✗ |
| KG-GPT(Kim et al., 2023a)[EMNLP23] | ✓ | ✗ | ✗ | ✗ |
| KB-BINDER(Li et al., 2023b)[ACL23] | ✗ | ✗ | ✗ | ✗ |
| ToG(Sun et al., 2024)[ICLR24] | ✗ | ✗ | ✗ | ✗ |
| RefKG (ours) | ✓ | ✓ | ✓ | ✓ |

In this paper, we introduce RefKG, an innovative framework specifically crafted to enhance the reasoning capabilities of LLMs through reflective engagement with knowledge graphs. In particular, RefKG is structured as a three-step framework: 1) A *Query Decoupling Module* that decouples a complex query into multiple sub-queries that share a common knowledge background. 2) A *LLM-Driven Knowledge Graph Exploration Module* that iteratively and reflectively retrieves relevant evidence subgraphs from a knowledge base and refines the knowledge through an expert model.. 3) An *Inference with Knowledge Reconstruction Module* that transforms structured knowledge into a natural language format that is more easily understood by the LLM, and integrates it with the question to derive the answer. Compared to approaches that directly use retrieved results in prompts (Kim et al., 2023a), our approach maximizes the reflection capabilities of LLMs (Asai et al., 2023) to critically assess and refine the evidence subgraph. Furthermore, we have formulated a knowledge-driven multi-task tuning strategy that provides RefKG with foundational expertise in knowledge-intensive reasoning. This is achieved by fine-tuning the model on a specially synthesized corpus, equipping it with the necessary skills for advanced reasoning tasks. Together, the three-step process enables our approach to autonomously retrieve, reflect, and utilize knowledge in solving knowledge-intensive tasks. In summary, our main contributions are three-fold:

- We propose RefKG, an innovative framework specifically crafted to enhance the reasoning capabilities of LLMs through reflective engagement with knowledge graphs. In particular, our approach simplifies complex queries through decomposition, enabling effective retrieval, reflection and reasoning within knowledge graphs.
- We develop an LLM-Generated corpus for knowledge-intensive multi-task tuning, equipping LLMs with initial expertise in knowledge-intensive reasoning, setting the stage for more advanced task-specific learning.
- We extensively evaluate RefKG on fact verification and knowledge graph question answering tasks. The experimental results affirm that RefKG not only outperforms previous KG-augmented methods across various open-source LLMs but also enhances the explainability of LLMs' reasoning process.

## 2 RELATED WORK

### 2.1 KG RETRIEVAL-AUGMENTED METHODS

Knowledge graphs (KGs) organize relationships between entities in a structured manner. Using KG retrieval to enhance LLMs has proven effective in alleviating hallucination phenomena (Agrawal et al., 2023; Pan et al., 2023). Recently, there has been increasing research on KG retrieval, which can be broadly categorized into two methods: (1) *Semantic Parsing-Based Methods*: These methods encode input questions, transforming them into logical forms (LF), and then execute retrieval on the KG. For example, SSKGQA (Li & Ji, 2022) generates query graphs based on natural language questions, attempting to leverage the semantic structure of questions to eliminate incorrect query graph structures. RnG-KBQA (Ye et al., 2022) introduces a ranking-and-generating framework that ranks candidate LF from questions and generates the final LF based on the top-k candidates. However, these approaches require generating executable SPARQL statements and additional label information. (2) *Information Retrieval-Based Methods*: PullNet (Sun et al., 2019) extracts a subgraph of the KG starting from topic entities and employs Graph Neural Networks to predict entity answers within the subgraph. UniK-QA (Oguz et al., 2022), based on the Seq2Seq framework, retrieves triplets from the knowledge graph and combines them with the question to generate answers. However, while this method depends on the accuracy of the retrieved subgraphs or triplets, it lacks mechanisms to filter

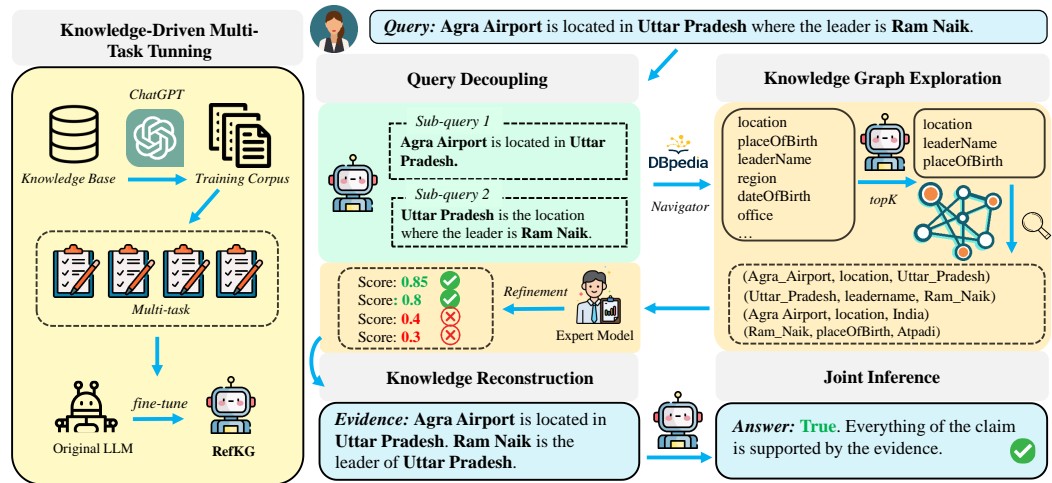

Figure 1: An overview of our proposed framework RefKG. The framework consists of three modules: *Query Decoupling*, *Evidence Subgraph Retrieval* and *inference with Knowledge Reconstruction*. We enhance the model's ability to utilize knowledge through Knowledge-Driven Multi-Task Tuning, enabling the decoupling, navigation, refinement, and reconstruction of knowledge.

or sift through the retrieval results, which can readily result in the accumulation of errors. DiFaR (Baek et al., 2023a) enhances retrieval accuracy by exploiting the representation similarity between queries and triplets but struggles with complex multi-hop QA problems. In contrast, our approach fully leverages the powerful semantic capabilities of LLMs to guide the knowledge graph retrieval process and applies reasonable quality control to the retrieval results.

## 2.2 LLMs Reasoning for KGQA

Recently, an increasing amount of work has been dedicated to leveraging the reasoning abilities of LLMs in Knowledge Graph Question Answering (KGQA). KAPING (Baek et al., 2023b) and KG-GPT (Kim et al., 2023a) prompts LLMs to generate answers by placing retrieved triplets into pre-defined prompts. However, they overlook the fact that triplets retrieved from KGs may not be in the form of natural language text, potentially increasing the difficulty of LLMs' inference. Moreover, they all use APIs providing by the black box model for inference and cannot train and deploy the model. Retrieve-Rewrite-Answer (Wu et al., 2023) addresses this issue by introducing a pre-training task and fine-tuning open-source LLMs on KG-to-text corpora, transforming triplet form text into free form text that is more easily comprehensible for LLMs. Nevertheless, the aforementioned methods do not filter the extracted triplets. This oversight may lead to irrelevant information being input into LLMs, thereby misleading them into generating wrong results.

## 3 Methodology

As shown in Figure 1, our proposed RefKG is structured with three modules: Query Decoupling, LLM-Driven Knowledge Graph Exploration, Inference with Knowledge Reconstruction.

### 3.1 Query Decoupling

When processing multi-hop queries, challenges like extended reasoning paths, semantic drift, and information loss are frequently encountered. Inspired by the divide-and-conquer paradigm, we initially decouple a complex query into multiple sub-queries, each of which shares the contextual semantics but contains only a single-hop atomic query. Specifically, given a knowledge-intensive query denoted as $q$, alongside a collection of knowledge entities $E$ related to this question, the objective is to predict the hop number $H$, derive a sequence of sub-queries $q_{sub} = [q_1, ..., q_H]$, and to ascertain the corresponding entity subset $E_{sub} = [e_1, ..., e_H]$ that are pertinent to these sub-queries.

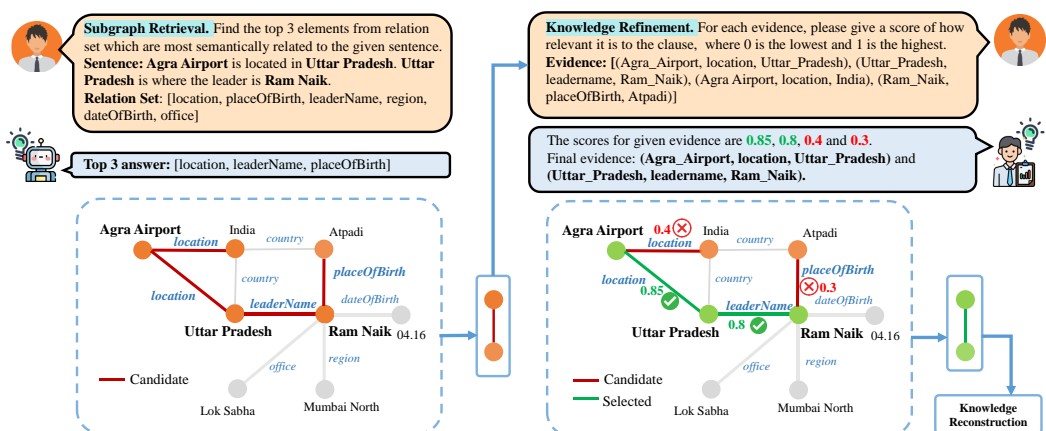

Figure 2: In the Evidence Subgraph Retrieval process, RefKG initiates from entities within the related subgraph to select the most probable relations, thereby constructing an inference pathway in triplets-form. In the Knowledge Refinement phase, RefKG uses a trained expert model to score and rerank the retrieved knowledge, filtering out noisy triplets.

In knowledge-intensive tasks, we assume entities contain the essential information necessary for the decomposition process. By anchoring entities, LLMs can capture the underlying mechanisms of knowledge-intensive problem decoupling. More precisely, given a complex query $q$, a set of entities $E$, and a predefined decoupling template $P$, the objective of query decoupling is to decouple the original query into several sub-queries $q_i$ and their corresponding entity subsets $e_i$, formulated as:

$$\{q_i, e_i\}_{i=1}^H = \mathbf{LLM}(p'), \quad p' = \mathbf{P}(q, e), \tag{1}$$

where $H$ represents the number of sub-queries after decoupling, which also typically corresponds to the number of hops in the original query. The notation $p'$ represents the sequence obtained by filling the query $q$ and the entity set $e$ into the slots of a predefined prompt template $P$.

As shown in Figure 1, the input text is:

 $q$: Agra Airport is located in Uttar Pradesh where the leader is Rant Naik.
 $e$: {Agra Airport, Uttar Pradesh, Rant Naik}

The LLM predicts a hop number of 2 and decoouples it into two parts:

 $q_1$:Agra Airport is located in Uttar Pradesh.
 $e_1$:{Agra Airport, Uttar Pradesh}.
 $q_2$:Uttar Pradesh is the location where the leader is Rant Naik.
 $e_2$:{Uttar Pradesh, Rant Naik}.

## 3.2 LLM-Driven Knowledge Graph Exploration

The evidence subgraph retrieval consists of *Evidence Subgraph Retrieval* and *Knowledge Refinement*.

**Evidence Subgraph Retrieval.** As shown in Figure 2, our approach leverages the LLM as a navigator, encouraging it to autonomously select the search trajectory on the related subgraph $\mathcal{G}_{sub}$, continuously advancing to form a chain of reasoning. Specifically, we divide the retrieval reasoning process into multiple iterations, ultimately forming a complete chain $\mathcal{P}_t$, formulated as:

$$\mathcal{P}_t = \{(e_1^{head}, r_1, e_1^{tail}) \xrightarrow{LLM} \dots \xrightarrow{LLM} (e_t^{head}, r_t, e_t^{tail}), (e_t^{head}, r_t, e_t^{tail}) \in \mathcal{G}_{sub}\} \tag{2}$$

For each iteration, the LLM conducts interpretable reasoning on the graph by targeting relationships as objectives for selecting paths. We formulate the relation selection task as an optimization problem, with the objective of maximizing the probability of extracting a set of relationships $r$ from the

knowledge graph $\mathcal{G}$ by generating an inference path $\mathcal{P}_t$:

$$P_\theta(r|q,e,\mathcal{G}) = \sum_{p_{t-1}\in\mathcal{P}_{t-1}} P_\theta(r|p_{t-1},q,e,\mathcal{G})P_\theta(p_{t-1}|q,e,\mathcal{G}), \tag{3}$$

where $p_{t-1}$ denote the historical reasoning path, $\theta$ denotes the parameters of LLMs, $P_\theta(p_{t-1}|q,e,\mathcal{G})$ is the probability of generating historical reasoning path given the query $q$, entity $e$ and knowledge graph $\mathcal{G}$, and $P_\theta(r|p_{t-1},q,e,\mathcal{G})$ indicates the probability of generating the relationship set $r$ through the query $q$, entity $e$ knowledge graph $\mathcal{G}$ and historical reasoning path $p_{t-1}$.

The new relation $r$ are incorporated into the reasoning path to form new reasoning paths $p_t$, with $N$ such paths together constituting a complete evidence subgraph $\mathcal{G}_{evi} = \{p_t^n\}_{n=1}^N$. To enhance the stability and coverage of relation selection, our approach does not incorporate only a single relation into the reasoning chain, but rather combines the Top-k most relevant relations.

**Evidence Subgraph Refinement.** The evidence subgraph retrieved from large knowledge graphs through complex searches inevitably contains noisy knowledge, which may severely impede the accuracy of factual reasoning. Therefore, refining the evidence subgraph becomes essential. To this end, we trained an expert model to refine and rerank the initially generated evidence subgraph, enhancing the accuracy and effectiveness of the external knowledge.

However, employing LLMs directly to generate knowledge scores presents challenges in terms of stability and accuracy. Transforming the generative task into a direct scoring task can significantly enhance the consistency and accuracy of the results. By minimizing the randomness in the generation process, the evaluation becomes more reliable.

In this approach, given a sub-query $q_i$ and its corresponding evidence subgraph $\mathcal{G}_{sub,i} \in \mathcal{G}_{sub}$, we utilize an LLM to jointly encode the query and subgraph together, resulting in a hidden layer state $h_i$. Then we integrate a single Multi-Layer Perceptron (MLP) after an LLM for regression training, aimming to map the hidden layer state $h_i$ to a corresponding score $s_i$ (more details in Appendix A.7). The formula for this mapping is expressed as follows:

$$h_i = \mathbf{LLM}(q_i, \mathcal{G}_{sub,i}), \quad s_i = \mathbf{MLP}(h_i) \tag{4}$$

We use the Mean Squared Error (MSE) loss as the objective function, formulated as:

$$\mathrm{MSE} = \frac{1}{n}\sum_{i=1}^n (s_i - \hat{s}_i)^2 \tag{5}$$

where $n$ is the number of samples, $s_i$ represents the actual scores, and $\hat{s}_i$ denotes the predicted scores by the model. Then, we rerank the obtained evidence triplets by score and set a threshold $\alpha$ to filter out triplet reasoning paths that are irrelevant to the question.

## 3.3 INFERENCE WITH KNOWLEDGE RECONSTRUCTION

To enhance the LLM's ability to profoundly understand external knowledge, our approach reconstructs the evidence subgraph into a natural language format. In this process, the LLM performs implicit logical reasoning and supplements the structured knowledge with additional content.

For an evidence subgraph $\mathcal{G}_{evi}$ containing $n$ triplets, we first linearize it into textual triplet form by connecting the head entity, relation, and tail entity.

We transform them into a textual prompt $p'$ by a predefined template $P$, and input $p'$ into a trained LLM to generate the textual evidence $evi$:

$$evi = \mathbf{LLM}(p'), \quad p' = \mathbf{P}(\mathcal{G}_{evi}), \quad G_{evi} = \{(e_n^{head}, r_n, e_n^{tail})\}_{n=1}^N \tag{6}$$

We perform reasoning in two types of knowledge-intensive tasks. For question answering tasks, the LLM is tasked with generating answers that most accurately align with the question, drawing upon the provided evidence. For fact verification tasks, the LLM assesses the evidence to ascertain whether it substantiates or contradicts the initial statement. We unify them into a single probabilistic model, formulated as:

$$P_\theta(a|q,\mathcal{G}) = P_\theta(a|evi,q,\mathcal{G})P_\theta(evi|q,\mathcal{G}) \tag{7}$$

where $a$ denote the answer, $evi$ denote the evidence transformed from knowledge graph. And the details of prompts templates for each step of RefKG are thoroughly outlined in Appendix A.12.

### 3.4 KNOWLEDGE-DRIVEN MULTI-TASK TUNNING

#### 3.4.1 TRAINING CORPUS

**Corpus Generation.** Addressing the gap left by existing corpora, which do not meet our specific training needs, we have initiated a multi-task-driven strategy for corpus generation. Recognizing ChatGPT's exceptional abilities in understanding and generating text, as highlighted in recent research (Li et al., 2023a; Tahmid Rahman Laskar et al., 2023), we have employed ChatGPT as a corpus generator. To create training data, we focus on three specific tasks: (1) Query Decoupling, (2) Evidence Subgraph Retrieval, and (3) Inference with Knowledge Reconstruction. For each task, we have pre-defined a template $T$, into which we insert feature elements $x$ relevant to each task, forming a text prompt $p$. This prompt $p$ is then fed into ChatGPT, generating the corresponding training corpora $y$. See Appendix A.4 for more details.

**Quality Control.** In light of the lack of explicit labels and the challenge of applying general metrics, we have developed specific evaluation methods for assessing the quality of the generated outcomes: (1) *Query Decoupling*: We evaluate the decoupling quality based on the entity set $E$ extracted from the original question and the entity sets $E_{div} = [e_{div,1}, ..., e_{div,H}]$ derived from the decomposed sub-queries. The criteria for considering the decoupling results as high-quality are as follows: (a) $E \neq \emptyset$. (b) $E = \bigcup_{i=1}^{H} e_{div,i}$. (c) If $|E_{\text{div}}| > 1$, then $\forall e_{div,i} \in E_{\text{div}}, e_{div,i} \subsetneq E$. If $|E_{\text{div}}| = 1$, then $E_{\text{div}} = E$. (2) *Inference with Knowledge Reconstruction*: We perform a unified assessment of the two-step pipeline process. For the answers $A$ generated through these two steps, we identify instances where the feedback from the generator corresponds with the factual ground truth as indicators of high-quality data.

#### 3.4.2 TRAINING

Previous research has explored the conditions under which multi-task learning is effective, concluding that it performs well when tasks are diverse yet related (Ni et al., 2023). To this end, we have designed tasks that are related in terms of knowledge background but involve significant differences in skills.

In the training phase, we synergistically infuse both linguistic and entity knowledge into LLMs by focusing on the optimization of three tasks: (1) *Query Decoupling* enables the model to learn how to decouple multi-hop queries into single-hop sub-queries. (2) *Evidence Subgraph Retrieval* enhance the model's ability to navigate, search, and generate evidence subgraphs within a knowledge graph. (3) *Evidence Generation* enhances the model's ability to translate evidence from the graph's structural format into coherent textual form, bridging the gap between graph-based data and natural language, and reasoning based on the evidence to derive answers.

The auto-regressive training objective focuses on training the LLM to predict subsequent tokens based on previous tokens. Specifically, for the prompt $p_i$ of different tasks, the objective function for generating the target answer $z = [z_1, ..., z_T]$ is:

$$\mathcal{L}_i(\theta) = -\sum_{t=1}^{T} \log p_\theta(z_t | \boldsymbol{z}_{<t}, \boldsymbol{p}_i) \tag{8}$$

where $\theta$ represents all the parameters of the model, and the prompt contains demonstration examples for different tasks: for query decoupling, $p_{dec}$ is filled with the query $q$ and entity set $e$. For knowledge reconstruction, $p_{rec}$ incorporates the evidence subgraph $\mathcal{G}_{sub}$. For Joint Inference, $p_{inf}$ incorporates the evidence $evi$ and the query $q$.

## 4 EXPERIMENTS

### 4.1 DATASETS

We evaluate RefKG on a fact-verification benchmark: FactKG, and two KGQA benchmarks: MetaQA and WebQSP. FactKG and WebQSP are both highly challenging benchmarks, while MetaQA is relatively less difficult. We also conducted experiments on these tasks to validate the broad adaptability of our approach across different domain contexts.

Table 2: Performance of different models on the FactKG benchmark. Performance marked with ∗ are sourced from (Kim et al., 2023b) and those marked with † are sourced from (Kim et al., 2023a). We applied our method, RefKG, to experiments on four open-source large language models(Baichuan-2, Llama-2, Internlm-2, Bloom), testing it against five types of questions (One-hop, Conjunction, Existence, Multi-hop, Negation). The best-performing results are in bold.

| Method | One-hop | Conjunction | Existence | Multi-hop | Negation | Overall |
|---|---|---|---|---|---|---|
| *Claim Only* | | | | | | |
| BERT∗ | 69.64 | 63.31 | 61.84 | 70.06 | 63.62 | 65.20 |
| BlueBERT∗ | 60.03 | 60.15 | 59.89 | 57.79 | 58.90 | 59.93 |
| Flan-T5∗ | 62.17 | 69.66 | 55.29 | 60.67 | 55.02 | 62.70 |
| Baichuan-2 7B | 29.88 | 26.21 | 18.55 | 18.43 | 17.73 | 24.29 |
| Llama-2 7B | 13.17 | 2.58 | 20.40 | 10.08 | 24.35 | 9.64 |
| Internlm-2 7B | 39.98 | 40.54 | 28.71 | 48.00 | 34.55 | 40.40 |
| Bloom 7B | 3.24 | 16.61 | 2.16 | 13.80 | 7.69 | 10.37 |
| *With Evidence* | | | | | | |
| KG-GPT†[EMNLP23] | - | - | - | - | - | 72.68 |
| GEAR∗[ACL19] | 83.23 | 77.68 | **81.61** | 68.84 | 79.41 | 77.65 |
| *RefKG (ours)* | | | | | | |
| Baichuan-2 7B | 81.14 | 83.75 | 80.83 | 73.52 | 77.63 | 80.30(+2.65) |
| Llama-2 7B | 84.13 | **88.46** | 72.83 | 71.83 | **83.64** | 81.26(+3.61) |
| Internlm-2 7B | 84.18 | 86.12 | 76.15 | 76.41 | 80.06 | 82.04(+4.39) |
| Bloom 7B | **85.65** | 87.94 | 81.14 | **77.81** | 82.80 | **84.04**(+6.39) |

**FactKG** is a fact-verification benchmark based on KG, containing 108K natural language statements verifiable via DBpedia (Lehmann et al., 2015), categorized into five reasoning types: One-hop, Conjunction, Existence, Multi-hop, and Negation. Furthermore, FactKG contains various patterns, including colloquial style statements as well as written style statements, to increase practicality.

**WebQuestionsSP** is a KGQA benchmark containing full semantic parses in SPARQL queries for 4,737 questions(3,098 train, 1,639 test). It is built on Freebase and includes multi-hop questions, linked through topic entities, reasoning chains, and SPARQL queries. It provides semantic parses in SPARQL with standard Freebase entity identifiers, which can be directly executed on Freebase to return answers to questions.

**MetaQA** is a comprehensive benchmark for assessing question-answering systems, particularly those utilizing knowledge graphs. It comprises over 400K questions, including one-hop, two-hop, and three-hop reasoning. This dataset is crucial for evaluating knowledge graph-based question answering, especially in handling complex multi-hop reasoning and noisy inputs.

## 4.2 IMPLEMENTATION DETAILS

We perform our experiments across a diverse range of LLMs, including Llama-2 7B (Touvron et al., 2023), Bloom 7B (Workshop et al., 2022), Baichuan-2 7B (Yang et al., 2023) and Internlm-2 7B (Team, 2023). For evidence subgraph retrieval, we configure the number of relations $k$ to be either 2 or 5. For knowledge refinement, we establish a score threshold $\alpha$ of 0.6. All our experiments are carried out on a NVIDIA 8×A800-SXM4-80G machine. See Appendix A.6 for more details.

## 4.3 BASELINES

**FactKG.** We compare RefKG with two types of baselines: (1) *Claim Only*: These baselines utilize the claim as the input without any evidence retrieved from the knowledge graph, including classifiers trained on the training set such as BERT, BlueBERT, and popular LLMs. (2) *With Evidence*: These baselines incorporate both the claim and retrieved evidence as inputs. This group includes fully supervised models like GEAR (Zhou et al., 2019) and 12-shot model KG-GPT (Kim et al., 2023a).

**MetaQA & WebQSP.** We compare RefKG with four types of baselines: 1) *Embedding-based methods*. 2) *Retrieve-augmented methods*. 3) *Prompting-based LLMs methods*, and 4) *Fine-tuned LLMs methods*. The details of each baseline are described in Appendix A.5.

Table 3: The performance of the models on WebQSP. The best results are in bold.

| Method | Hits@1 |
|---|---|
| *Embedding* | |
| EmbedKGQA(Saxena et al., 2020)[ACL20] | 66.6 |
| NSM(He et al., 2021)[WSDM21] | 68.7 |
| TransferNet(Shi et al., 2021)[EMNLP21] | 71.4 |
| *Retrieval* | |
| GraftNet(Sun et al., 2018)[EMNLP18] | 66.4 |
| PullNet(Sun et al., 2019)[EMNLP19] | 68.1 |
| SR+NSM(Zhang et al., 2022)[ACL22] | 68.9 |
| *LLM (Prompting)* | |
| KAPING(Baek et al., 2023b)[NLRSE23] | 73.9 |
| KB-BINDER(Li et al., 2023b)[ACL23] | 74.4 |
| ChatGPT+ToG(Sun et al., 2024)[ICLR24] | 76.2 |
| GPT4+ToG(Sun et al., 2024)[ICLR24] | 82.6 |
| *LLM (Fine-tuned)* | |
| InstructGraph(Yu et al., 2023)[ACL24] | 73.3 |
| UniKGQA(Jiang et al., 2022)[ICLR23] | 77.2 |
| Retrieve-Rewrite(Wu et al., 2023)[IJCKG23] | 79.4 |
| DECAF(Yu et al., 2023)[ICLR23] | 82.1 |
| RefKG (ours) | **85.2** |

Table 4: The performance of the models on MetaQA (Hits@1). The best results are in bold.

| Methods | 1-hop | 2-hop | 3-hop | Avg. |
|---|---|---|---|---|
| *Embedding* | | | | |
| KVMemNN(Xu et al., 2019)[NAACL19] | 96.2 | 82.7 | 48.9 | 75.9 |
| EmbedKGQA(Saxena et al., 2020)[ACL20] | 97.5 | 98.8 | 94.8 | 97.0 |
| NSM(He et al., 2021)[WSDM21] | 97.1 | **99.9** | 98.9 | 98.6 |
| *Retrieval* | | | | |
| GraftNet(Sun et al., 2018)[EMNLP18] | 97.0 | 94.8 | 77.7 | 89.9 |
| PullNet(Sun et al., 2019)[EMNLP19] | 97.0 | **99.9** | 91.4 | 96.1 |
| *LLM (Prompting)* | | | | |
| ChatGPT | 60.0 | 23.0 | 38.7 | 40.6 |
| KG-GPT(Kim et al., 2023a)[EMNLP23] | 96.3 | 94.4 | 94.0 | 94.9 |
| StructGPT(Jiang et al., 2023)[EMNLP23] | 97.1 | 97.3 | 87.0 | 93.8 |
| KB-BINDER[ACL23] | 93.5 | 99.6 | 96.4 | 96.5 |
| *LLM (Fine-tuned)* | | | | |
| UniKGQA(Jiang et al., 2022)[ICLR22] | 97.5 | 99.0 | **99.1** | 98.5 |
| Retrieve-Rewrite(Wu et al., 2023)[IJCKG23] | - | 97.7 | - | 97.7 |
| RefKG (ours) | **98.1** | 99.4 | 99.0 | **98.8** |

## 4.4 MAIN RESULTS

**Results on FactKG.** The results are shown in Table 2. We can make the following observations:

First, RefKG with Bloom-7B outperforms all baseline methods in terms of the overall accuracy, attaining a new state-of-the-art status on this benchmark. This success can be attributed to our framework's dual strategy of employing knowledge graphs as external resources and harnessing the innate reasoning powers of LLMs. By encouraging LLMs to engage deeply with and reflect on retrieved information, RefKG significantly enhance the performance.

Second, fine-tuned 7B-parameter LLMs exhibit much better performance in fact verification tasks than LLMs without fine-tuning. Notably, RefKG enhances the performance of Llama 2, Bloom, Internlm 2 and Baichuan 2 by 71.62%, 73.67%, 41.64% and 56.01%, respectively.

Third, in the context of knowledge graph retrieval methods, RefKG outperforms KG-specific supervised models like GEAR and training-free approaches such as KG-GPT. This underscores the effectiveness of our approach, which involves fine-tuning LLMs with a rich set of instructions. Moreover, RefKG demonstrates commendable results across all five tasks, with the exception of the Existence category. This exception might stem from the limited entity information available, which poses challenges for effective query decoupling.

**Results on WebQSP.** The results are shown in Table 3. RefKG demonstrates competitive performance, achieving a Hits@1 score of 85.2% within fine-tuned LLMs methods. Moreover, unlike prompting-based LLMs methods that typically rely on carefully crafted prompts to guide black-box large models in generating answers, RefKG surpasses them by fine-tuning a 7B-parameter LLM.

**Results on MetaQA.** The results are shown in Table 4. Notably, RefKG reaches state-of-the-art performance on the Hop-1 test set, recording a 98.1% accuracy. This exceptional performance is attributed to the richer relational context available in Hop-1 compared to Hop-2 and Hop-3, suggesting that the strategic use of LLMs for relation selection significantly minimizes errors at this juncture, thereby enhancing overall results. Additionally, RefKG achieves performances close to SOTA on the Hop-2 and Hop-3 test sets, underscoring its versatility and robust adaptability across a variety of tasks. This demonstrates RefKG's consistent and reliable performance across both single-hop and multi-hop question answering tasks.

## 4.5 ABLATION STUDY

As illustrated in Table 5, we perform a series of ablation experiments on FactKG.

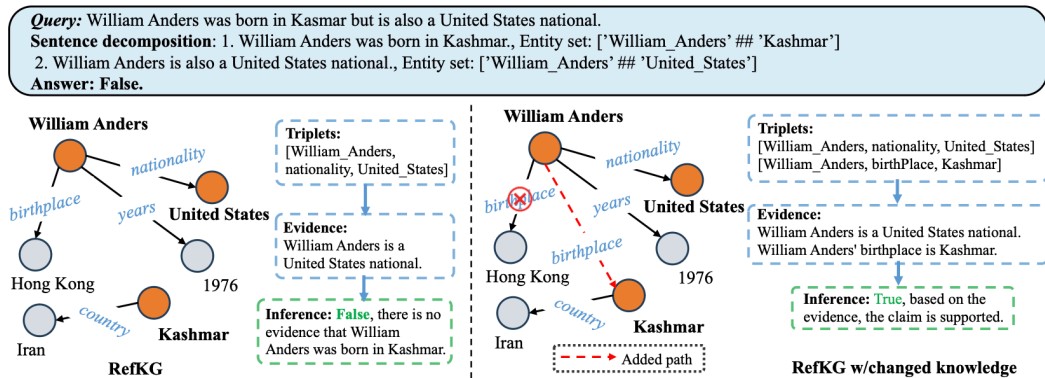

Figure 3: A case study on FactKG. The left figure illustrates the process of RefKG handling a claim, while the right figure depicts the modification made to the knowledge graph, resulting in the change of RefKG's response.

**Is text form better than triplet form?** In our exploration, we experiment with changing the format of evidence provided to the LLM and observe that substituting natural language text with triplets led to a decline in performance by approximately 20%. This observation leads us to believe that triplets may omit essential semantic details, complicating the model's ability to process the information effectively. Conversely, presenting evidence in natural language form appears to be more in harmony with the LLM's pre-training corpus format, thereby enhancing the model's capacity to assimilate and utilize the evidence more efficiently.

**What role does multi-task tuning play in RefKG?** In our investigation, we experiment with substituting the instruction fine-tuning models with *llama 2* at specific stages within RefKG and observe a marked decline in performance. We find that models without fine-tuning cannot grasp the question-answer patterns in the prompts well, often providing vague or invalid answers. To further validate the effectiveness of multi-task tunning, we conducted additional comparative experiments on four LLMs. Specifically, we train the LLM on multiple independent single tasks and then combine the trained LLMs into a unified system for inference. We refer to this approach as single-task tuning. As shown in Figure 4, single-task tuning weakens the model's overall capabilities compared to multi-task tuning, leading to a decline in task performance, with an average accuracy drop of 3.21%.

We propose that multi-task tuning provides three significant benefits : (1)Multi-task tuning enables the model to share hidden layers across different tasks, thereby facilitating the sharing of learned features and representations. This mechanism assists the model in leveraging knowledge acquired from one task to enhance performance on other tasks. (2)Training sereval tasks simultaneously can reduce the risk of overfitting and enhance the model's generalization ability. (3)Multi-task tuning improves data utilization efficiency, reduces the consumption of computational resources, and decreases complexity during deployment.

Additionally, we explore the potential of Lora fine-tuning, which, while beneficial, does not achieve the same level of impact as multi-task instruction fine-tuning.

**How does Knowledge Refinement enhance inference performance?** By eliminating Knowledge Refinement in the RefKG framework, we observe a noticeable decline in overall performance by about 2.71%. This observation underscores the value of reflection in the model's analytical process. Through reflection, the model can autonomously filter out incorrect relations and evidence on a semantic level. This strategic narrowing of focus is crucial for improving the accuracy of the model's inferences.

Table 5: Ablation study on the FactKG. "Rate" quantifies the reduction in accuracy. JI tunning denotes the Joint Inference tunning.

| Methods | Accuracy | Rate |
|---|---|---|
| **RefKG (full)** | **81.26** | 0.00 |
| *-triplet only* | 61.15 | -20.11 |
| *-w/o Knowledge Refinement* | 78.55 | -2.71 |
| *-w/o Knowledge Reconstruction* | 68.99 | -12.27 |
| *-w/o JI tunning* | 50.62 | -30.64 |
| RefKG (Lora-ft) | 72.45 | -8.81 |

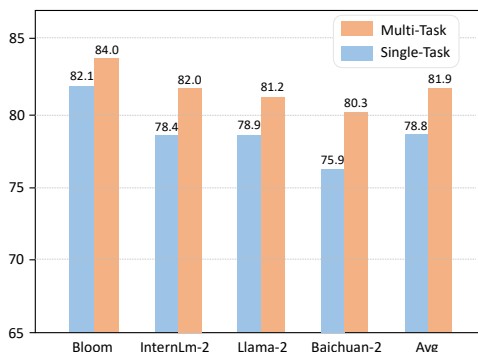

Figure 4: Comparison of Multi-task Tuning and Single-task Tuning on Bloom, InternLM-2, Llama-2, and Baichuan-2.

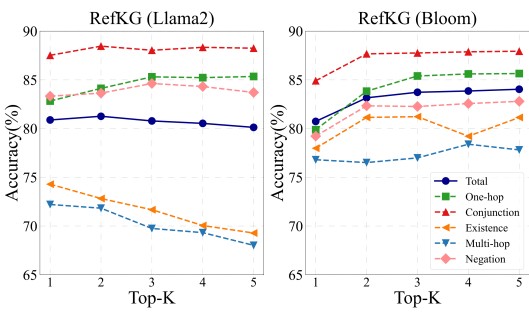

Figure 5: Impact of varying the number of Top-K retrieval with Llama-2 and Bloom on FactKG.

### 4.6 FURTHER ANALYSIS

**Impact of numbers of Top-K Retrieval.** As shown in Figure 5, we explore Top-K values ranging from 1 to 5 and observed a notable trend: distinct LLMs exhibit optimal performance at varying Top-K values. Bloom model demonstrates a relatively consistent performance across different tasks, showing a trend of improvement as the Top-K value increased, but Llama model experiences a progressive decrease in performance with higher Top-K settings. This trend suggests that an increase in the number of paths selected, and consequently, more evidence being generated, may overwhelm the Llama model, complicating its ability to distill crucial information from an extensive pool of evidence. Interestingly, setting the Top-K value to 1, where only the most probable relation is chosen from the set, RefKG still delivered commendably high performance. This observation suggests that the singular relation selected by the LLM is often the correct one, highlighting the models' capability to pinpoint relevant information even within a constrained selection scope.

**Qualitative Analysis** We conduct a case study as presented in Figure 3. Based on the given statement, our method RefKG performs sentence decomposition to identify triplets and transform them into evidence. Since no relevant fact is found in the knowledge graph for the statement *"William Anders was born in Kashmar"*, our model outputs *"False"*. This underscores RefKG's capability to precisely detect the absence of supporting evidence for incorrect statements and to consequently deliver an accurate verdict. Furthermore, we explore whether RefKG can adapt to newly updated knowledge. By manually adding a path into the original KG, our model adeptly identifies and processes the triplets into evidence, resulting in a diametrically opposed conclusion. This case demonstrates the model's ability to seamlessly adjust to updated factual knowledge, negating the necessity for further training or adjustments. This flexibility highlights RefKG's potential for maintaining relevance and accuracy in the face of evolving knowledge bases.

## 5 CONCLUSION

In this paper, we proposed the RefKG framework, which engages with knowledge graphs in a reflective manner to identify the most likely relational paths and evidence, using this curated evidence to derive answers. To Infuse the LLM with the abilities to decouple, navigate, refine, reconstruct, and reason over knowledge, we developed a knowledge-driven multi-task tuning approach and built a corresponding training corpus. The experimental results prove its effectiveness on fact verification and knowledge graph question answering. Our method can be deployed on any open-source LLM, and the experimental results indicate that it achieves excellent performance in two knowledge-intensive tasks: fact verification and knowledge graph question answering.

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

## A APPENDIX

### A.1 LIMITATIONS

In this section, we faithfully discuss the limitations of our approach and potential avenues for future research.

**Cumulative error effect**. Although our framework can handle some complex multi-hop or negation questions, it involves multiple subtasks. The workflow of the pipeline generates a cumulative error effect. For example, if the model misidentifies entities in the first step of sentence decomposition, subsequent answers obtained will inevitably be incorrect. Therefore, future work could focus on reducing error rates by introducing efficient and accurate retrieval methods or instruction fine-tuning methods.

**Larger model sizes**. Limited by computational resources, we only applied RefKG to the 7B LLM and conducted full-parameter fine-tuning of the model under this configuration without testing larger models. We hope to conduct experiments on models with larger parameter sizes such as OPT (175B) in the future.

### A.2 LARGE LANGUAGE MODELS

We conducted extensive experiments on multiple 7B open-source LLMs, including popular models such as *Llama-2*, *Bloom*, *Vicuna*, *Internlm-2* and *Baichuan-2*.

**Llama-2** is an LLM optimized for dialogue scenarios based on Llama 2, particularly suitable for handling KGQA tasks. *Vicuna* is an LLM fine-tuned based on Llama 1.

**Bloom** is an LLM trained on the Megatron-LM GPT2, utilizing unique decoder structures, normalization of the word embedding layer, linear bias attention position encoding with the GeLU activation function, and other advanced techniques.

**Baichuan-2** is developed by Baichuan Intelligence, is a highly influential AI large-scale model. It integrates intent understanding, information retrieval, and reinforcement learning technologies, achieving high-performance results through supervised fine-tuning and alignment with human intent.

**Internlm-2** is capable of efficiently supporting ultra-long contexts of up to 200,000 characters, achieving a leading level among open-source models in tasks such as Longbench and E-eval. Its comprehensive capabilities have shown all-around advancements over the previous generation of Internlm, and it possesses strong code interpretation and data analysis abilities.

### A.3 DATASETS

We conduct extensive experiments on three datasets: FactKG (Kim et al., 2023b), MetaQA (Zhang et al., 2018) and WebQuestionsSP (Yih et al., 2016).

**FactKG**   is a fact-verification benchmark based on KG, containing 108K natural language statements verifiable via DBpedia (Lehmann et al., 2015), categorized into five reasoning types: One-hop, Conjunction, Existence, Multi-hop, and Negation.Furthermore, FactKG contains various linguistic patterns, including colloquial style statements as well as written style statements, to increase practicality.

**MetaQA**   is a comprehensive benchmark for assessing question-answering systems, particularly those utilizing knowledge graphs. It comprises over 400K questions, including one-hop, two-hop, and three-hop reasoning. This dataset is crucial for evaluating knowledge graph-based question answering, especially in handling complex multi-hop reasoning and noisy inputs.

**WebQuestionsSP**   is a KGQA benchmark containing full semantic parses in SPARQL queries for 4,737 questions(3,098 train, 1,639 test). It is built on Freebase and includes multi-hop questions, linked through topic entities, reasoning chains, and SPARQL queries. It provides semantic parses in SPARQL with standard Freebase entity identifiers, which can be directly executed on Freebase to return answers to questions.

## A.4 CORPUS GENERATION

We use the GPT-3.5-turbo API (\$0.002 / 1K tokens) to generate training corpora, with the following steps:

**Query Decoupling.** Given a question q and a set of entities e, we insert them into a predefined generation template $p_{dec}$ to obtain a text prompt. This text prompt is then input into ChatGPT to produce an output sequence $z = [z_1, ..., z_T]$, which includes sub-queries and their respective entity subsets.

**Knowledge Reconstruction.** Given an evidence subgraph $\mathcal{G}evi$ stored in triplet form, we first linearize it into a text format by concatenating the head entity, relation word, and tail entity to form textual triplets. We insert this sequence of triplets into a predefined template $p_{evi}$: "Your task is to transform a knowledge graph in triplets (or tuples) format into a single sentence, preserving the original words or expressions from the triplets as much as possible. The knowledge graph is: graph. The sentence is:". This prompt is then fed into ChatGPT, resulting in an output sequence $z = [z_1, ..., z_T]$ that contains the textualized evidence.

**Joint Inference.** Given a query $q$ and an evidence sequence $evi$, we insert both into a predefined template $p_{inf}$, input it into ChatGPT, and the model will produce inference results and explanations based on the input.

## A.5 BASELINES

We compare RefKG with four types of baselines: 1) *Embedding-based methods*. 2) *Retrieve-augmented methods*. 3) *Prompting-based LLMs methods*, and 4) *Fine-tuned LLMs methods*. The details of each baseline are described below.

**Embedding-based methods**

- KVMemNN (Xu et al., 2019) utilizes a Key-Value memory network to store triples and conducts multi-hop reasoning through iterative operations on the memory.
- EmbedKGQA (Saxena et al., 2020) approaches reasoning on knowledge graphs as a sequential link prediction problem by leveraging the embeddings of both entities and questions.
- NSM (He et al., 2021) employs a sequential model to replicate the multi-hop reasoning process.
- TransferNet (Shi et al., 2021) uses a graph neural network to capture the relevance between entities and questions for reasoning. process.

**Retrieve-augmented methods**

- GraftNet (Sun et al., 2018) retrieves relevant subgraphs from knowledge graphs using entity linking.
- PullNet (Sun et al., 2019) trains a retrieval model that combines an LSTM and a graph neural network to retrieve a question-specific subgraph.
- SR+NSM (Zhang et al., 2022) introduces a relation-path retrieval mechanism to retrieve subgraphs for multi-hop reasoning.

**Prompting-based LLMs methods**

- KB-Binder (Li et al., 2023b) is the first to enable few-shot in-context learning over KBQA tasks.
- KAPING (Baek et al., 2023b) propose to augment the knowledge directly in the input of LLMs.

Table 6: Hyper-parameters of training.

| Hyper-parameters | FactKG | MetaQA | WebQSP |
|---|---|---|---|
| training strategy | full | full | lora |
| epoch | 3 | 3 | 50 |
| sequence length | 2048 | 256 | 2048 |
| learning rate | 1e-5 | 2e-5 | 5e-5 |
| batch size | 1 | 1 | 16 |
| gradient accumulation | 4 | 4 | 1 |
| optimizer | AdamW | AdamW | AdamW |
| weight decay | 0.01 | 0.01 | 0.01 |
| deepSpeed stage | 3 | 3 | 3 |

- KG-GPT (Kim et al., 2023a) is a multi-purpose framework leveraging LLMs for tasks employing KGs. It comprises three steps: Sentence Segmentation, Graph Retrieval, and Inference, each aimed at partitioning sentences, retrieving relevant graph components, and deriving logical conclusions, respectively.

- StructGPT (Jiang et al., 2023) proposes an invoking linearization-generation procedure to support LLMs in reasoning on the structured data.

- ToG (Sun et al., 2024) enables LLM agent to interactively explore related entities and relations on KGs and perform reasoning based on the retrieved knowledge.

**Fine-tuned LLMs methods**

- KD-CoT (Wang et al., 2023) retrieves pertinent knowledge from knowledge graphs to formulate faithful reasoning plans for LLMs.

- UniKGQA (Jiang et al., 2022) integrates graph retrieval and reasoning into a unified model with LLMs, achieving state-of-the-art performance on KGQA tasks.

- DECAF (Yu et al., 2023) synergizes semantic parsing and LLMs reasoning to jointly generate answers, achieving notable performance on KGQA tasks.

- Retrieve-Rewrite-Answer (Wu et al., 2023) propose an answer-sensitive KG-to-Text approach that can transform KG knowledge into well-textualized statements most informative for KGQA. A. Also, they propose a KG-to-Text enhanced LLMs framework for solving the KGQA task.

- InstructGraph (Wang et al., 2024) is a framework that empowers LLMs with the abilities of graph reasoning and generation by instruction tuning and preference alignment.

A.6    IMPLEMENTATION DETAILS

The details of training hyperparameters are presented in Table 6.

**FactKG.**    We extracted a subset of 40,000 data from the training set to generate our training corpus. Following quality control measures, we produced a total of 86,786 data instances, divided into three categories: 31,999 for question decomposition, 29,702 for evidence generation, and 24,085 for evidence reasoning. For the task of evidence subgraph retrieval, we configure the number of relations $k$ to be either 2 or 5, and the score threshold $\alpha$ to 0.6. For a full-parameter fine-tuning of a 7b model using two A800-80G graphics cards, the memory consumption is approximately 140G, and it takes about 24 hours.

**MetaQA.**    We extracted a subset of 30,000 data from the training set to create our training corpus. In the hyperparameter configuration, we set the number of selected relations k to 3, and the score threshold $\alpha$ to 0.7. Since the number of entities related to each question in the WebQSP dataset is smaller compared to FactKG, we directly treat the topic entity as the sole member of the entity set, in order to train the LLM's ability to predict the number of hops. For a full-parameter fine-tuning of a 7b model using two A800-80G graphics cards, the memory consumption is approximately 140G, and

Table 7: Statistics on 100 incorrect samples.

| Stage | Total | Existence | Multi-hop | Other |
|---|---|---|---|---|
| Query Decoupling | 62 | 10 | 18 | 34 |
| Evidence Subgraph Retrieval | 13 | 7 | 1 | 5 |
| Joint Inference | 25 | 6 | 3 | 16 |

it takes about 16 hours.

**WebQuestionsSP.** We first extract SPARQL queries and their corresponding topic entities from the training set. Next, we parse these SPARQL queries and decompose them into multiple hops. By designing precise SPARQL query statements, we perform searches in Freebase, thereby obtaining inference chains represented in the form of triplets. By populating predefined task templates with the obtained ground truth data, we construct training datasets for each stage. And we set the number of selected relations k to 3, and the score threshold $\alpha$ to 0.6. Since the number of entities related to each question in the WebQSP dataset is smaller compared to FactKG, we directly treat the topic entity as the sole member of the entity set, in order to train the LLM's ability to predict the number of hops. Due to the small size of our training dataset, which contains only 3,098 entries, we use Lora for fine-tuning to prevent overfitting during the training process. For a lora fine-tuning of a 7b model using four A800-80G graphics cards, the memory consumption is approximately 240G, and it takes about 14 hours.

## A.7 TRAINING DETAILS FOR EXPERT MODEL

We trained the Expert LLM using 30,000 annotated data entries, as detailed below:

**Evidence score annotation.** For each sub-query $q_i$ and triplet format evidence $t$, we first employ the semantic similarity model DistilBERT to assign a similarity score, denoted as $s$, to represent the supportiveness of the evidence triplet toward the query.

For each sub-query $q_i$, we sort all evidence triplets $t$ based on their scores, from highest to lowest. This set includes triplets that are relevant to the query as well as some that are noise. We then match these triplets with the ground truth. If a triplet from the ground truth is ranked among the top $k$, we retain it as part of the training data; otherwise, we filter it out.

It's important to note that we don't directly use all the collected annotated data for training. Instead, we first conduct a complete inference process with this data. If the final inference result is correct, we retain the annotated data as the gold score; if it's incorrect, we discard it. This approach ensures the high quality of the annotated data. Additionally, to minimize the influence of noise during the training process, we have eliminated anomalously high and low scores.

## A.8 ERROR ANALYSIS.

For the error analysis of the FactKG, see Table 7.

To explore the execution efficiency of each step, we perform an error analysis on FactKG. It was noted that errors predominantly arise in the Query Decoupling stage, primarily due to the model's struggle in correctly identifying entities within sentences, a difficulty that is particularly pronounced in Multi-hop claims. This issue can lead to the alteration of entities mentioned in a sentence. A potential solution to mitigate such errors involves enhancing the model's sensitivity towards entity recognition.

## A.9 NOISE ANALYSIS.

We randomly selected 100 samples from the FactKG dataset and conducted a detailed analysis of the noise introduction and reduction in the decoupling, retrieval, scoring, and reconstruction steps, as shown in Table 8. We defined three statistical metrics:

Table 8: Noise analysis of the FactKG.

| Type | Decoupling | Retrieval | Refinement | Reconstruction |
|---|---|---|---|---|
| Noise Introduction | 9 | 24 | 2 | 3 |
| Noise Reduction | - | - | 16 | 6 |
| Correctness | 92 | 87 | 85 | 84 |

- **Noise introduction**: Refers to the introduction of incorrect knowledge, conflicting knowledge, or loss of correct information at a particular step.

- **Noise reduction**: Refers to successfully removing incorrect or irrelevant knowledge at a particular step.

- **Correctness**: Indicates whether the current knowledge information contains correct knowledge.

The details of the noise flow in the four stages are as follows:

**In the Query Decoupling**: A small number of cases may experience partial entity information loss, leading to the introduction of noise.

**In the Subgraph Retrieval**: Since we aim to retrieve as much relevant knowledge as possible, it is inevitable to introduce some irrelevant knowledge and even knowledge that conflicts with correct information. Among them, some conflicting information may interfere with the results, while some irrelevant information has a minor impact.

**In the Knowledge Refinement**: Some incorrect and irrelevant triples are scored and removed, but there is also a small possibility that a few correct answers may be mistakenly filtered out.

**In the Knowledge Reconstruction**: While converting triples into textual information, the model performs implicit reasoning. During this process, the model may actively discard some incorrect or conflicting information and even correct erroneous information, but this may also result in the loss of a few correct pieces of information.

The statistical results show that noise introduction is often difficult to completely avoid when handling complex problems. Through the collaborative operation of various tasks, particularly during the **Knowledge Refinement** and **Knowledge Reconstruction** stages, we effectively control noise, significantly mitigating its cumulative effects across tasks and reducing its impact on overall performance. This further validates the robustness and effectiveness of our approach in complex knowledge reasoning scenarios.

### A.10    MORE DETAILS ABOUT ABLATION STUDY.

We conducted additional comparison experiments using an untrained base model to complete the entire process. In the experiments, Base Model represents the results obtained by directly using the untrained model, while RefKG represents the results achieved by applying our method. The experimental results are shown in the Table 9.

The experimental results show that directly using the untrained Base Model leads to an average performance drop of 46.11%. This indicates that untrained models struggle to handle our designed multi-task framework and are limited in their ability to tackle tasks involving complex knowledge.

This demonstrates that RefKG significantly enhances the model's adaptability and performance through carefully designed tasks and targeted training. Our task design emphasizes knowledge reconstruction, refinement, and joint inference, with these steps working collaboratively to form a comprehensive reasoning mechanism. This enables the model to better handle complex knowledge scenarios and question-answering tasks.

Table 9: More Details About Ablation Study.

| Model | Base Model | RefKG (ours) | Difference |
|---|---|---|---|
| Llama-2 7B | 34.12 | 81.26 | -47.14 |
| Bloom 7B | 37.65 | 84.04 | -46.39 |
| Internlm-2 7B | 39.41 | 82.04 | -42.63 |
| Baichuan-2 7B | 31.73 | 80.30 | -48.57 |
| Average | 35.73 | 81.84 | -46.11 |

Table 10: Evaluation of computational efficiency.

| Dataset | Number of triplets | Number of calls | Inference time |
|---|---|---|---|
| FactKG | 10.11 | 4.8 | 2.4 |
| WebQSP | 19.76 | 4.4 | 2.1 |
| MetaQA | - | 5.1 | 1.9 |
| Average | - | 4.8 | 2.1 |

## A.11 EVALUATION OF COMPUTATIONAL EFFICIENCY

We randomly selected 100 samples from each of the three datasets for the efficiency analysis, as shown in Table 10. We computed the average number of triples involved in each question, the average number of LLM calls, and the average inference time (in seconds).

## A.12 PROMPTS

The 9-shot prompt templates for Query Decoupling, Evidence Subgraph Retrieval, and Joint Inference are respectively presented in Table 11, Table 12, and Table 13.

## A.13 QUALITATIVE RESULTS

More qualitative results on FactKG and MetaQA are respectively presented in Table 14 and Table 15.

---

**Prompt for query decoupling**

---

Please decompose the given sentence into multiple single-hop sub-sentences, which can be represented by a triplet. Each entity subset should contain no more than two elements, entities can be duplicated across different subsets, and the union of multiple subsets should equal the original entity set. Generate the results in the format of (number). (Sentence), (entity set), using "##" to separate different entities. Refer to the following examples to complete the task:

Examples)
Sentence A: The City of Soldotna is the owner of the AIDAluna.
Entity set: ['AIDAluna' ## awareaware'"The City of Soldotna"']
Answer: 1. The City of Soldotna is the owner of the AIDAluna., Entity set: ['AIDAluna' ## '"The City of Soldotna"']

Sentence B: Born in Gevelsberg, Alan Shepard was awarded the "Distinguished Service Medal".
Entity set: ['Alan_Shepard' ## 'Distinguished_Service_Medal_(United_States_Navy)' ## 'Gevelsberg']
Answer: 1. Alan Shepard was awarded the "Distinguished Service Medal"., Entity set: ['Alan_Shepard' ## 'Distinguished_Service_Medal_(United_States_Navy)'] 2. Alan Shepard was born in Gevelsberg., Entity set: ['Alan_Shepard' ## 'Gevelsberg']

......
Your Task)
Query: ««QUERY»»
Entity set: ««ENTITY_SET»»
Answer:

---

Table 11: Prompt for query decomposition. ««QUERY»» and ««ENTITY_SET»» will be replaced with the corresponding query and entity set in the FactKG dataset.

---

**Prompt for evidence subgraph retrieval**

---

I will give you a set of words.

Find the top ««K»» elements from relational words set which are most semantically related to the given sentence. You may select up to ««K»» words. If there is nothing that looks semantically related, pick out any ««K»» elements and give them to me.

Examples)
Sentence A: The City of Soldotna is the owner of the AIDAluna.
Words set: ['status', 'owner', 'builder', 'shipOwner', 'shipBuilder', 'operator', 'shipOperator', 'shipClass']
Top 2 Answer: ['owner', 'shipOwner']
Sentence B: Born in Gevelsberg, Alan Shepard was awarded the "Distinguished Service Medal".

Relational words set: ['birthPlace', 'mission', 'awards', 'rank', 'region', 'state', 'birthYear', 'country', 'type']
Top 2 Answer: ['birthPlace', 'awards']
... Now let's find the top ««K»» elements.
Query: ««QUERY»»
Relational words set: ««RELATION_SET»»
Top ««K»» Answer:

---

Table 12: Prompt for subgraph retrieval. ««<QUERY»» and ««ENTITY_SET»» will be replaced with the corresponding query and Relational words set in the FactKG dataset. ««K»» will be replaced with the chosen hyperparameter $k$.

---

**Prompt for joint inference**

---

You should verify the claim based on the textual evidence. Each evidence is derived from one or several sentences generated from knowledge graph triplets.
Verify the claim based on the evidence. (True means that everything contained in the claim is supported by the evidence.) Choose one of {True, False}, and give me one sentence explanation.

Examples)
Claim A: The City of Soldotna is the owner of the AIDAluna.
Evidence: Lack of evidence.
Answer: {False}, there is no evidence that The City of Soldotna is the owner of the AIDAluna.

Claim B: Brandon Carter was born in England and graduated from the University of Cambridge where the current Chancellor is Leszek Borysiewicz.
Evidence: Brandon Carter attended the University of Cambridge.Brandon Carter was born in England.Leszek Borysiewicz served as the Vice-Chancellor of the University of Cambridge.
Answer: {True}, everything of the claim is supported by the evidence.

Now let's verify the Claim based on the Evidence.
Query: ««QUERY»»
Evidence: ««EVIDENCE»»
Answer:

---

Table 13: Prompt for joint inference.««QUERY»» and ««EVIDENCE»» will be replaced with the corresponding query on the FactKG dataset and evidence set generated in 3.3.

| Type | Claim | Evidence Subgraph Graph | Textual Evidence generation | Prediction |
|---|---|---|---|---|
| One-hop | Do you know Agra Airport IATA Location Identifier is AGR. | [Agra_Airport, iataLocationIdentifier, "AGR"], | Agra Airport has an IATA location identifier of "AGR". | True |
| Conjunction | Doris Bures is the leader of Austria where Alfons Gorbach died in Styria. | [Austria, leader, Doris_Bures], [Alfons_Gorbach, placeOfDeath, Styria], [Doris_Bures, birthPlace, Austria] | Austria is the leader and birthplace of Doris Bures. Alfons Gorbach was born and died in Styria. | True |
| Existence | At least Dawn Butler had a successor! | [Dawn_Butler, successor, Paul_Boateng], [Dawn_Butler, birthPlace, England], [Dawn_Butler, predecessor, Sarah_Teather] | Dawn Butler has a successor named Paul Boateng. Dawn Butler was born in England. Dawn Butler has a predecessor named Sarah Teather. | True |
| Negation | I understand that Acura is not a division of Honda. | [Acura, owningCompany, Honda], [Honda, division, Acura], [Acura, owner, Honda] | Acura is owned by Honda and is also a division of Honda. | False |
| Multi-hop | It is located in Alan B Miller Hall in the United States. | [Alan_B_Miller_Hall, location, Williamsburg,_Virginia], [Williamsburg,_Virginia, country, United_States] | Alan B Miller Hall locates in Williamsburg,Virginia. Williamsburg,Virginia is in the United States. | True |

Table 14: Qualitative results from FACTKG.

| Task | Question | Evidence Subgraph Graph | Textual Evidence generation | Prediction |
|---|---|---|---|---|
| 1-hop | Brian Backer appears in which movies? | (Moving Violations, starred_actors, Brian Backer) | Moving Violations, starring Brian Backer, is a film. | Moving Violations |
| | who is the writer of the film Habit? | (Habit, written_by, Larry Fessenden) | Habit, written by Larry Fessenden. | Larry Fessenden |
| | what kind of film is The Old Dark House? | (The Old Dark House, has_genre, Comedy) (The Old Dark House, has_genre, Horror) | The Old Dark House is a Comedy. The Old Dark House is a Horror genre. | Comedy |
| 2-hop | what genres do the movies written by Anders Nilsson fall under? | (The Third Wave, written_by, Anders Nilsson), (The Third Wave, has_genre, Action) | The Third Wave, written by Anders Nilsson, is an action genre. | Action |
| | what are the primary languages in the movies directed by David Mandel | (EuroTrip, directed_by, David Mandel) (EuroTrip, in_language, German) | EuroTrip, directed by David Mandel, is a film in the German language. | German |
| | who is listed as director of Joseph Stein written films | (Fiddler on the Roof, written_by, Joseph Stein), (Fiddler on the Roof, written_by, Joseph Stein) | Fiddler on the Roof, written by Joseph Stein and directed by Norman Jewison, is a film. | Norman Jewison |
| 3-hop | what genres do the films that share writers with Karate-Robo Zaborgar fall under? | (Karate-Robo Zaborgar, written_by, Noboru Iguchi), (RoboGeisha, written_by, Noboru Iguchi), (RoboGeisha, has_genre, Action) | Karate-Robo Zaborgar and RoboGeisha are written by Noboru Iguchi and they both belong to the genre of Action. | Action |
| | the movies that share writers with the movie Naqoyqatsi were released in which years? | (Naqoyqatsi, written_by, Godfrey Reggio), (Powaqqatsi, written_by, Godfrey Reggio), (Powaqqatsi, release_year, 1988) | Naqoyqatsi and Powaqqatsi were written by Godfrey Reggio and were released in 1988. | 1988 |
| | who is listed as screenwriter of the movies directed by the The Battle of Shaker Heights director? | (The Battle of Shaker Heights, directed_by, Kyle Rankin), (Infestation, directed_by, Kyle Rankin), (Infestation, written_by, Kyle Rankin) | The Battle of Shaker Heights and Infestation, directed by Kyle Rankin, were written by Kyle Rankin. | Kyle Rankin |

Table 15: Qualitative results from MetaQA.