# OpenReview forum: "Reflection on Knowledge Graph for Large Language Models Reasoning"
_ICLR.cc/2025/Conference — Submitted to ICLR 2025_

### Official Review · Reviewer_gUWH · 2024-10-22

**Soundness:** 3
**Presentation:** 3
**Contribution:** 2
**Rating:** 6
**Confidence:** 4

**Summary:**

The paper introduces RefKG, a new framework designed to enhance the reasoning capabilities of Large Language Models (LLMs) by improving their integration with knowledge from knowledge graphs. RefKG tackles the problems of noise and error accumulation in knowledge retrieval and reasoning processes, which previously impeded the effective use of external knowledge in answering complex questions. The framework employs a four-step process: decomposing complex queries, retrieving and pruning knowledge graphs to form evidence subgraphs, generating textual evidence, and performing evidence-enhanced reasoning. RefKG also incorporates a multi-task tuning strategy that not only feeds knowledge into LLMs but also trains them on effectively utilizing this information for question answering. Experimental results on tasks such as fact verification and knowledge graph question answering demonstrate that RefKG outperforms existing state-of-the-art models, indicating a significant improvement in LLMs' ability to handle knowledge-intensive tasks.

**Strengths:**

* The idea of using query decoupling, knowledge retrieval, and reconstruction is sound, and the paper is well written.
* The experiments are clean, and the figures in the paper are easy to follow.

**Weaknesses:**

W1: The idea of using query decoupling, knowledge retrieval, reconstruction is sound but not that interesting. Seems each step has already been acknowledged by other papers which may underscore the novelty of the proposed unified framework. Especially as the statement from lines 144-145, "the aforementioned methods do not filter the extracted triplets..." further underestimates the contribution compared to methods like Retrieve-Rewrite-Answer, KAPING. The issue of filtering extracted triplets has already been discussed in papers like ToG[2], FiDeLIS[3], etc. If the authors intend to discuss the solutions to this issue, the corresponding references should be involved.

W2: the table 1 comparison seems not fair, seems KG-GPT, KB-BINDER, and TOG are all training-free methods, only Retrieve-rewriter methods require training. However, the training phase in Retrieve-rewriter is only targeted to train the retriever and rewriter, where the target is different from the proposed RefKG. In that case, it's not very sound to compare properties like multi-task tuning and knowledge refinement. Otherwise, I suggest the authors should also consider more baselines requiring further training like RoG[1].

W3: the training process is quite similar to the existing method like RoG[1] and there is no comparison and analysis between these papers. Additionally, I'm quite curious whether the training process is necessary; it seems like the proposed method can be independent only with inference. In that case, considering adding another ablation study is necessary, especially using some advanced models like GPT-4o or o1. (btw, what is the model used in Table 5 ablation study?)


References:

* [1] Reasoning on graphs: Faithful and interpretable large language model reasoning (https://arxiv.org/pdf/2310.01061)
* [2] Think-on-Graph: Deep and Responsible Reasoning of Large Language Model on Knowledge Graph
* [3] FiDeLiS: Faithful Reasoning in Large Language Model for Knowledge Graph Question Answering

**Questions:**

Q1: What are the reflection capabilities explicitly referred to in lines 72 to 73? I suggest the authors should rephrase the definition of reflection of LLMs in case not all readers are familiar with the term in the context of LLMs.

Q2: How is the corresponding entity subset $E_{sub}$ collected as mentioned in Section 3.1? How to make sure the decoupled entities can be grounded to the corresponding KGs? If the subsets are provided in the dataset like FactKG, how is that method been adapted to dataset like WebQSP?

Q3: How to define the ending point of the chain $P_t$ mentioned in Section "Evidence subgraph retrieval"? Is the entire process controlled by the LLM itself?

Q4: What are the LLMs used across the entire method section? Have the LLMs been fine-tuned using the corpus mentioned in Section 3.4, or only the naive LLMs? Additionally, what is the LLM used for in the expert model mentioned from lines 238 to 250?

Q5: I have concerns about whether the knowledge reconstruction process may inadvertently introduce noise/hallucinations when leveraging LLMs to transform the retrieved KG triplets into some textual statements. Since this process is not under control and perhaps requires some curated designs or error analysis.

---

> ### Author Response · Authors · 2024-11-22
> **Response to Reviewer gUWH （1/3）**
>
> Thank you for your valuable comments. We will explain your concerns point by point.
>
>
>
> **Comment 1:**
>
> What are the reflection capabilities explicitly referred to in lines 72 to 73?
>
> **Reply:**
>
> Reflection capabilities specifically refer to the comprehensive ability of large language models to handle complex knowledge scenarios by leveraging multi-task collaboration for deep analysis and decoupling of input content, semantic understanding, noise detection, and implicit reasoning. We approach this from two perspectives.
>
> **Multi-task Collaborative Capabilities**: In our framework of Decoupling-Exploration-Refinement-Reconstruction-Inference, reflection capabilities are demonstrated through the dynamic collaboration and deep reasoning of the model across various tasks.
>
> - In the **Query Decoupling** phase, the LLM breaks down multi-dimensional complex questions into single-hop atomic problems, reducing problem complexity and improving the precision of knowledge matching.
> - In the **Subgraph Retrieval** phase, the LLM leverages its reasoning capabilities to search for knowledge in the knowledge graph relevant to each subquery.
> - In the **Knowledge Refinement** phase, the model filters and assigns weights to evidence, identifying noisy information and prioritizing knowledge that better supports the query.
> - In the **Knowledge Reconstruction** phase, the model leverages implicit reasoning to reorganize triples into natural language information that better aligns with the context. This collaborative mechanism across tasks significantly enhances the overall reasoning depth and robustness of the model.
>
> **Deep Knowledge Reasoning Ability**: Through our knowledge-driven multi-task instruction fine-tuning method, the model goes beyond shallow understanding and generation of input content, acquiring reflection capabilities based on knowledge graphs. Specifically, after training, the model can deeply evaluate the plausibility of information, actively identify and correct potential erroneous knowledge. This capability surpasses the traditional generative mode of LLMs, enabling the model to perform deep reasoning in complex knowledge scenarios.
>
>
>
> **Comment 2:**
>
> How is the corresponding entity subset Esub collected as mentioned in Section 3.1? How to make sure the decoupled entities can be grounded to the corresponding KGs? If the subsets are provided in the dataset like FactKG, how is that method been adapted to dataset like WebQSP?
>
> **Reply:**
>
> For the FactKG dataset, since the entity set is already provided, we leverage the entity set during the "Query Decoupling" stage to assist the LLM in efficiently performing question decoupling. In contrast, the WebQSP dataset has lower question complexity and fewer related entities compared to FactKG.
>
> Therefore, we designate the topic entity as the sole member of the entity set, serving as the starting point for the first sub-query to train the LLM's ability to predict the number of hops.  Testing results show that the LLM achieves a hop number prediction accuracy of exceeding 97%, highlighting its high effectiveness in this task.

---

> > ### Author Response · Authors · 2024-11-22
> > ****Response to Reviewer gUWH （2/3）****
> >
> > **Comment 3:**
> >
> > How to define the ending point of the chain Pt mentioned in Section "Evidence subgraph retrieval"? Is the entire process controlled by the LLM itself?
> >
> > **Reply:**
> >
> > The entire process of "Evidence subgraph retrieval" is fully controlled by the trained LLM. Specifically, in the preceding step, "Query Decoupling," the LLM decomposes a complex question into multiple sub-queries, each of which can be represented as a triple, effectively forming a single-step query.
> >
> > Therefore, the total number of sub-queries corresponds to the number of hops N, which limits the number of iterations in the search process.
> >
> > During retrieval, the LLM begins with the topic entity and, guided by the current hop's sub-query, automatically selects up to  k  relationships from the candidate relations retrieved from the knowledge graph to obtain the tail entity. The search concludes after completing up to N hops, thereby forming a complete logical chain.
> >
> >
> >
> > **Comment 4:**
> >
> > What are the LLMs used across the entire method section? Have the LLMs been fine-tuned using the corpus mentioned in Section 3.4, or only the naive LLMs? Additionally, what is the LLM used for in the expert model mentioned from lines 238 to 250?
> >
> > **Reply:**
> >
> > Our method, RefKG, has been implemented on four open-source LLM models: Llama-2 7B, Baichuan-2 7B, InternLM-2 7B, and Bloom 7B.
> >
> > For each LLM model, we utilized a knowledge-driven multi-task fine-tuning corpus mentioned in Section 3.4 for training and evaluation without introducing any additional base models.
> >
> > Taking Llama-2 7B as an example, we performed knowledge-driven multi-task instruction fine-tuning on it. To ensure consistency, the expert model was also based on Llama-2 7B and specifically trained for expert scoring capabilities, thereby maintaining the entire process on the same open-source LLM model. The same approach was applied to the other three open-source LLMs as well.
> >
> > **Comment 5:**
> >
> > Analysis of noise introduced in the knowledge reconstruction.
> >
> > **Reply:**
> >
> > We rigorously selected high-quality training data to ensure that the model's knowledge reconstruction capabilities are thoroughly trained.
> >
> > For the knowledge reconstruction task, let $E$ represent the set of all entities in the evidence triples and $R$ represent the set of all relations in the evidence triples. If the reconstruction results fully contain $E$ and $R$, completeness is considered ensured. Additionally, by jointly reasoning with the textual evidence and the query, if the correct answer is obtained, correctness is considered ensured. Data that satisfies both **completeness** and **correctness** is regarded as high-quality.
> >
> > We randomly selected 100 samples for case analysis:
> >
> > - **Noise introduction**: The introduction of incorrect knowledge or the loss of correct information.
> > - **Noise reduction**: The successful removal of incorrect or irrelevant knowledge.
> > - **Correctness**: Whether the current knowledge information contains all correct knowledge.
> >
> > |        | Correctness | Noise Introduction | Noise reduction |
> > | :----: | :---------: | :----------------: | :-------------: |
> > | FactKG |     84      |         3          |        6        |
> > | WebQSP |     87      |         2          |        4        |
> >
> > The statistical results indicate that the noise introduced during the knowledge reconstruction phase is minimal and manageable. Moreover, the model's implicit reasoning during the generation of textual evidence effectively reduces part of the noise. This demonstrates that the knowledge reconstruction task achieves efficient information integration during the generation process, thereby improving the overall reasoning accuracy to a certain extent.

---

> > > ### Author Response · Authors · 2024-11-22
> > > **Response to Reviewer gUWH （3/3）**
> > >
> > > **Comment 6:**
> > >
> > > Demonstrating the effectiveness of the training process through comparison and analysis, as well as comparisons with other baselines.
> > >
> > > **Reply：**
> > >
> > > As shown in Table 5, we present the results of ablation experiments using Llama-2 on the FactKG dataset. By gradually removing individual tasks and observing the performance changes, we identified the following patterns:
> > >
> > > 1. **Removing the Knowledge Reconstruction task**: Directly reasoning with triples led to a **20.11%** performance decrease.
> > > 2. **Retaining the Knowledge Reconstruction task without training it**: Performance decreased by **12.27%**.
> > > 3. **Retaining the Knowledge Refinement task without training it**: Performance decreased by **2.71%**.
> > > 4. **Retaining the Joint Inference task without training it**: Performance decreased by **30.64%**.
> > >
> > > These results clearly demonstrate the significant contribution of each task to the overall performance improvement.
> > >
> > > In addition, we conducted new comparative experiments using an untrained model to complete the entire process. In the experiments, **Base Model** represents the results obtained by directly using the untrained model, while **RefKG** represents the results achieved by applying our method. The experimental results are as follows:
> > >
> > > |   Model    | Base Model | RefKG(ours) | Difference |
> > > | :--------: | :--------: | :---------: | :--------: |
> > > |  Llama-2   |   34.12    |    81.26    |   -47.14   |
> > > |   Bloom    |   37.65    |    84.04    |   -46.39   |
> > > | Interlm-2  |   39.41    |    82.04    |   -42.63   |
> > > | Baichuan-2 |   31.73    |    80.30    |   -48.57   |
> > > |  Average   | **35.73**  |  **81.84**  | **-46.11** |
> > >
> > > The experimental results show that directly using the untrained **Base Model** leads to an average performance drop of **46.11%**. This indicates that untrained models struggle to handle our designed multi-task framework and are limited in their ability to tackle tasks involving complex knowledge.
> > >
> > > In contrast, **RefKG** significantly enhances the model's adaptability and performance through carefully designed tasks and targeted training. Our task design emphasizes knowledge reconstruction, refinement, and joint inference, with these steps working collaboratively to form a comprehensive reasoning mechanism. This enables the model to better handle complex knowledge scenarios and question-answering tasks.
> > >
> > > We also provide a discussion with "Reasoning on Graphs: Faithful and Interpretable Large Language Model Reasoning" here.
> > >
> > > RoG primarily designed two instruction tuning tasks:
> > >
> > > - planning optimization : enable  LLMs to generate faithful relation paths as plans.
> > > - retrieval-reasoning optimization : enables LLMs to reason based on the retrieved reasoning paths.
> > >
> > > Compared to other methods, our RefKG approach has several distinct features:
> > >
> > > - We have incorporated a question decomposition mechanism as the first step, enabling the model to effectively handle structurally complex long sentences.
> > > - We trained a expert scorer based on LLMs that identifies and filters out noise triplets that do not support the answering of questions during the retrieval process, significantly enhancing the accuracy of reasoning tasks across various knowledge scenarios.
> > > - Our designed knowledge module effectively converts triplets into textual form, allowing the model to understand and process information more naturally and in-depth.
> > > - We constructed a multi-task instructional dataset and performed multi-task tuning on it to infuse knowledge into the large language model.
> > >
> > >
> > >
> > > We have revised the manuscript according to the Reviewer’s suggestion and response to each comment provided in the Weakness section above. We hope that our rebuttal aligns with the reviewer’s expectations, and we hope that the Reviewer can consider possibly giving a higher rating. Thanks.

---

> > > > ### Comment · Reviewer_gUWH · 2024-11-26
> > > >
> > > > Thank you for the detailed response. I suggest incorporating some of your rebuttals into the revised manuscript, especially the clarification in Comment 2 about the different experimental setup on FactKG and WebQSP. Overall, the method appears sound, though it still seems incremental, as what I mentioned in weakness 1, which lacks a direct response from the authors. Based on the clarifications provided, I will adjust my rating for the technique's integrity and look forward to further discussion with other reviewers and Area Chairs. Thanks.

---

### Official Review · Reviewer_7R7R · 2024-10-23

**Soundness:** 2
**Presentation:** 2
**Contribution:** 2
**Rating:** 5
**Confidence:** 4

**Summary:**

The paper introduces a framework called RefKG, designed to enhance the reasoning capabilities of LLMs by integrating them more effectively with KGs. The authors address the challenges faced by current approaches, which often introduce noise during knowledge retrieval and reasoning, leading to errors that hinder LLMs from effectively utilizing external knowledge for complex multi-hop questions.

RefKG operates through a three-step process:
- Query Decoupling Module: Decomposes complex queries into simpler sub-queries that share common knowledge backgrounds, facilitating more targeted retrieval.
- LLM-Driven Knowledge Graph Exploration Module: Iteratively and reflectively retrieves relevant evidence subgraphs from the knowledge base, using an expert model to refine the knowledge and eliminate irrelevant information.
- Inference with Knowledge Reconstruction Module: Transforms structured knowledge from the KG into natural language that the LLM can easily understand, integrating it with the original question to derive the answer.

Additionally, the authors develop a knowledge-driven multi-task tuning strategy by fine-tuning the LLM on a specially synthesized corpus generated by LLMs themselves. This equips the model with foundational expertise in knowledge-intensive reasoning, enhancing its ability to handle advanced tasks.
Experimental results on fact verification and KGQA tasks demonstrate that RefKG outperforms previous state-of-the-art models, not only improving performance but also enhancing the explainability of the LLMs' reasoning processes.

**Strengths:**

- The organization of the paper is clear and easy to follow, although there are some typos should be polished.
- RefKG presents an effective approach to integrating LLMs with KGs by leveraging reflective reasoning, addressing the limitations of previous methods. The framework's iterative retrieval and pruning effectively reduce noise in the retrieved knowledge, improving the accuracy of the reasoning process.
- The knowledge-driven multi-task tuning equips the LLM with initial expertise, improving its ability to handle knowledge-intensive tasks from the outset.
- The framework demonstrates superior performance on fact verification and KGQA tasks, validating its effectiveness over previous KG-augmented methods. Furthermore, RefKG is evaluated across various open-source LLMs, showing that it can be adapted to different models and settings.

**Weaknesses:**

- Although effective, the RefKG framework lacks technical novelty. The pipeline is simple and not exciting enough.
- The RefKG framework's effectiveness on tasks beyond fact verification and KGQA is not explored, limiting understanding of its broader applicability. Besides, the benchmarks are not sufficient enough.
- The approach may not generalize well to domains with sparse or highly specialized KGs. Moreover, the performance may heavily rely on the completeness and accuracy of the underlying KGs, which may vary in different domains.
- The iterative retrieval and reflection process may be computationally intensive, raising concerns about scalability for large-scale applications.
- The paper seems not go through a careful typos checking, as there are some typos.

**Questions:**

- Have you tested RefKG on other knowledge-intensive tasks or domains? If so, how did it perform compared to existing methods?
- How does RefKG perform in terms of computational efficiency, especially with large-scale knowledge graphs, and have you considered methods to optimize it?
- What steps were taken to identify and mitigate potential biases in the LLM-generated corpus used for multi-task tuning?

Missing References
- KnowledgeNavigator: Leveraging Large Language Models for Enhanced Reasoning over Knowledge Graph (Complex & Intelligent Systems, 2024)
- Chain-of-Knowledge: Integrating Knowledge Reasoning into Large Language Models by Learning from Knowledge Graphs (2024)
- Paths-over-Graph: Knowledge Graph Enpowered Large Language Model Reasoning (2024)
- LightRAG: Simple and Fast Retrieval-Augmented Generation (2024)
- ……

Tyops:
- At line 040, “like knowledge graphs (KGs)(Luo et al.,”, there is a missing blank between “(KGs)” and “(Luo et al.,”.
- In Table 1, “ToG(Sun et al., 2022)[ICLR24]”, the citation format of ToG should be latest, 2022 --> 2024.
- At line 207, Evidence Subgraph retrieval. --> Evidence Subgraph Retrieval.
- At line 506, Impact of numbers of Top-K retrieval. --> Impact of Numbers of Top-K Retrieval.
- ……

---

> ### Author Response · Authors · 2024-11-22
> **Response to Reviewer 7R7R （1/2）**
>
> Thank you for your valuable comments. We will explain your concerns point by point.
>
>
>
> **Comment 1：**
>
> The broad adaptability of RefKG in other knowledge-intensive tasks or domains.
>
> **Reply:**
>
> Our method, RefKG, focuses on question answering and fact verification, unifying these two tasks within a single modeling framework. This dedicated focus allows the framework to achieve optimal performance in these core tasks. However, other knowledge-intensive tasks, such as information extraction, knowledge graph construction, and knowledge graph completion, may require different optimization directions or modules, which are beyond the scope of our current research.
>
> We conducted experiments on both general-purpose and domain-specific datasets. FactKG and WebQSP utilize DBpedia and Wikidata, two large-scale knowledge graphs with extremely broad knowledge coverage, demonstrating RefKG's adaptability to complex question-answering tasks across multiple domains. Meanwhile, MetaQA, based on the MovieQA knowledge graph in the movie knowledge domain, further validates RefKG's exceptional performance in domain-specific tasks.
>
> We have summarized the knowledge graphs used in the three benchmark datasets, the corresponding knowledge domains, the number of triples required per query, the hop numbers of the queries, and the best performance of our method on these datasets, as detailed in the table below:
>
> | Benchmark |         Knowledge graph          | Domain  | Average number of triplets per query |  Hop num   | Accuracy |
> | :-------: | :------------------------------: | :-----: | :----------------------------------: | :--------: | :------: |
> |  FactKG   |  DBpedia (850 million triplets)  | General |                10.11                 |  1, 2, 3   |  84.04   |
> |  WebQSP   | Wikidata (1.57 billion triplets) | General |                19.76                 | 1, 2, 3, 4 |   85.2   |
> |  MetaQA   |     MovieQA（75k entities）      |  Movie  |                  2                   |  1，2，3   |   98.8   |
>
> It is worth emphasizing that our method incorporates knowledge-driven multi-task training, focusing on enhancing the model's ability to retrieve, refine, and apply knowledge, rather than limiting it to a specific knowledge domain. The capabilities learned by the model not only demonstrate broad generalization across various general-purpose domains but also support the plug-and-play integration of domain-specific knowledge graphs, enabling efficient performance on specialized tasks.
>
>
>
> **Comment 2：**
>
> What steps were taken to identify and mitigate potential biases in the LLM-generated corpus used for multi-task tuning?
>
> **Reply:**
>
> We have developed specific evaluation and filtering methods to monitor the quality of the generated corpus, as described in Section 3.4.1, "Quality Control".
>
> For the "Query Decoupling" task, let $E$ represent the entity set of the original sentence, and $E_{div,i}$ denote the entity set for each sub-query after decoupling. The criteria for considering the decoupling results as high-quality are as follows:
>
> (a) $E_{div} \neq \emptyset$.
>
> (b) $E = \bigcup_{i=1}^{H} e_{div,i}$.
>
> (c) If $\( |E_{\text{div}}| > 1 \)$, then $\( \forall e_{div,i} \in E_{\text{div}}, e_{div,i} \subsetneqq E \)$. If $\( |E_{\text{div}}| = 1 \)$, then $\( E_{\text{div}} = E \)$.
>
> For the knowledge reconstruction task, let $E$ represent the set of all entities in the evidence triples and $R$ represent the set of all relations in the evidence triples. If the reconstruction results fully contain $E$ and $R$, completeness is considered ensured. Additionally, by jointly reasoning with the textual evidence and the query, if the correct answer is obtained, correctness is considered ensured. Data that satisfies both completeness and correctness is regarded as high-quality.
>
> Additionally, the corpus we generate focuses on training models to utilize knowledge appropriately and does not involve content related to safety, ethics, or politics that may carry potential biases.

---

> > ### Author Response · Authors · 2024-11-22
> > **Response to Reviewer 7R7R （2/2）**
> >
> > **Comment 3：**
> >
> > The computational efficiency issues of RefKG, particularly the iterative retrieval and reflection process, involve significant computational overhead, raising concerns about scalability for large-scale applications.
> >
> > **Reply:**
> >
> > Firstly, our method RefKG employs a process of decoupling, retrieval, refinement, and reasoning to enable LLMs to engage in deep thought on knowledge graphs. By invoking the LLM multiple times to accomplish various tasks, this approach is not redundant but essential for fully tapping into the LLM's potential for deep understanding and utilization of knowledge, ensuring both accuracy of results and effective use of knowledge in a way that is irreplaceable.
> >
> > KAPING[1] and KB-BINDER[2] make only a few calls to the large language model (LLM), including just one instance. We conducted a comparison with KAPING on WebQSP (wikidata) as shown in the following graph:
> >
> > |     Method      |        Model Size         | number of calls | Accuracy |
> > | :-------------: | :-----------------------: | :-------------: | :------: |
> > |     KAPING      |           6.7B            |       few       |  53.34   |
> > |     KAPING      |           175B            |       few       |  69.58   |
> > |    KB-BINDER    | code-davinci-002(unknown) |       few       |   74.4   |
> > | **RefKG(ours)** |            7B             |    multiple     | **85.2** |
> >
> > The results indicate that the approach of making only a few calls to the LLM fails to fully exploit the potential of the LLM to solve complex problems, thus not achieving optimal performance.
> >
> > Secondly, we conducted a detailed quantification of the scale and difficulty of different tasks, as well as the average number of times RefKG invoked LLMs and the inference speed. We randomly selected 100 samples from each dataset for experimentation, and the results are shown in the table below:
> >
> > | Benchmark | Average triplets numbers | Average number of calls | Average inference time |
> > | :-------: | :----------------------: | :---------------------: | :--------------------: |
> > |  FactKG   |          10.11           |           4.8           |          2.4s          |
> > |  WebQSP   |          19.76           |           4.4           |          2.1s          |
> > |  MetaQA   |            -             |           5.1           |          1.9s          |
> > |  Average  |            -             |           4.8           |          2.1s          |
> >
> > Across the three datasets, the average number of LLM invocations was 4.8, and the average total inference time was 2.1 seconds.
> >
> > Finally, from a scalability perspective, RefKG employs knowledge-driven multitask instruction fine-tuning on LLMs, allowing a single LLM to exhibit multiple capabilities. With a one-time training and deployment, it can flexibly handle calls for various tasks. This approach not only conserves resources but also maintains the method's transferability and scalability.
> >
> > [1] Knowledge-Augmented Language Model Prompting for Zero-Shot Knowledge Graph Question Answering
> >
> > [2] Few-shot In-context Learning for Knowledge Base Question Answering
> >
> >
> >
> > **Comment 4:**
> >
> > About missing references and typos
> >
> > **Reply:**
> >
> > Thank you for pointing out the missing references and typographical errors in the paper. We sincerely apologize for our oversight. The missing references have been added, and all typographical errors have been corrected in the revised manuscript. We appreciate your thorough review and suggestions to improve the quality of our paper.
> >
> >
> >
> >
> >
> > We have revised the manuscript according to the Reviewer’s suggestion and response to each comment provided in the Weakness section above. We hope that our rebuttal aligns with the reviewer’s expectations, and we hope that the Reviewer can consider possibly giving a higher rating. Thanks.

---

> > > ### Comment · Reviewer_7R7R · 2024-11-25
> > >
> > > Dear Authors,
> > >
> > > Thank you very much for the clarification!
> > >
> > > After checking your response, I still have reservations about viewpoints of some weakness, especially the limited technical novelty, and decide to keep my original scores.
> > >
> > > Best Regards,
> > >
> > > Reviewer 7R7R

---

> ### Author Response · Authors · 2024-12-02
> **Replying to Reviewer 7R7R**
>
> We greatly appreciate you taking the time to review our work and provide valuable feedback. As the discussion phase between the authors and reviewers draws to a close, we would like to take this opportunity to further clarify our responses.
>
> In the latest version of the PDF, we have made several **modifications and additions**, including:
>
> - Corrected **tyops** and **unclear expressions**;
> - Added the **quantification and analysis of noise** in the overall process, further clarifying the impact of noise on model performance and providing a detailed explanation of the effectiveness of our noise control strategy;
> - Added an **additional comparative experiment**, which thoroughly analyzes the rationale behind our method design and training effectiveness, further validating the superiority of the method in complex tasks;
> - Introduced a **quantitative analysis of inference time**, demonstrating the efficiency of the model in large-scale applications.
>
> It is worth emphasizing that current approaches often introduce additional noise in the pipeline process of knowledge retrieval and reasoning, leading to the accumulation of errors, impeding LLMs from effectively combining the external knowledge in answering complex multi-hop questions. To this end, our method is specifically designed to enhance the reasoning capabilities of LLMs through reflective engagement with knowledge graphs, while effectively controlling the noise. We believe this method has tremendous potential to advance the field. Our main contributions are:
>
> - We introduce a knowledge-driven multi-task instruction fine-tuning method, which enables the model to effectively complete the full process of decoupling, exploration, refinement, reconstruction, and reasoning within a knowledge graph through multi-task collaboration. Multi-task fine-tuning allows the model to share learned features and representations across different tasks.
>
> - We trained an expert scoring model based on LLM. This module can identify and filter out triples that do not support question answering, effectively controlling the introduction of noise. This significantly enhances the accuracy of reasoning tasks across various knowledge scenarios. Additionally, this module improves the interpretability and efficiency of knowledge-based question-answering tasks.
>
> We believe this approach has great potential to advance the development of this field. Additionally, we have made appropriate revisions based on the feedback provided by the reviewers. We sincerely hope that our paper will be accepted.
>
> Once again, we sincerely thank you for your involvement and thoughtful feedback!

---

### Official Review · Reviewer_8UTd · 2024-11-02

**Soundness:** 3
**Presentation:** 3
**Contribution:** 2
**Rating:** 6
**Confidence:** 4

**Summary:**

The paper introduces RefKG, a framework that enhances LLMs' complex reasoning capabilities through reflective engagement with knowledge graphs. The framework consists of three main components: query decoupling, evidence subgraph retrieval, and knowledge reconstruction inference. Additionally, it employs a multi-task tuning strategy to improve LLMs' performance on knowledge-intensive tasks. The framework was evaluated on three benchmarks - FactKG, MetaQA, and WebQuestionsSP - demonstrating superior performance over previous state-of-the-art models in both fact verification and knowledge graph question answering tasks.

**Strengths:**

- The innovative approach of leveraging knowledge graphs through reflective reasoning significantly enhances LLMs' reasoning capabilities, particularly for complex multi-hop questions.
- The multi-task tuning strategy effectively expands LLMs' capabilities, showing substantial improvements across different tasks.
- The empirical evaluation is comprehensive, with thorough comparisons against various baseline models across multiple benchmarks.

**Weaknesses:**

- The error accumulation issue in the multi-step pipeline is not adequately addressed, potentially limiting the framework's effectiveness for more complex reasoning chains.
- The generalization capability across different domains is not thoroughly explored, lacking discussion on the framework's applicability to diverse knowledge domains.
- The interpretability aspects of RefKG, particularly regarding the decision-making process and reasoning paths during multi-task learning, could be better explained.

**Questions:**

This paper claims to address the noise and error accumulation issues in knowledge retrieval and reasoning pipelines. However, I have some concerns about this claim: (1) While decomposing complex queries into simpler sub-queries is interesting, this additional step could potentially introduce its own errors. The paper's ablation study shows that incorrect entity identification in this stage accounts for 62% of total errors. How do you ensure the reliability of this decomposition step, especially for queries with complex semantic dependencies? (2) The paper proposes using an expert model to score and filter evidence triplets. There's no analysis of how this refinement process handles conflicting or complementary evidence. (3) Can you provide quantitative analysis showing how the refinement process reduces noise propagation in multi-hop reasoning chains?

---

> ### Author Response · Authors · 2024-11-22
> **Response to Reviewer 8UTd （1/2）**
>
> Thank you for your valuable comments. We will explain your concerns point by point.
>
> **Comment 1:**
>
>  How do you ensure the reliability of this decomposition step, especially for queries with complex semantic dependencies?
>
> **Reply:**
>
> It is important to clarify that Table 7 presents the results of our traceability analysis on 100 error cases from the FactKG dataset. Due to the large number of entities and the high complexity of the questions in the FactKG dataset, the "Query Decoupling" task in the first step is relatively more challenging. Nevertheless, the accuracy of this task still exceeds **90%** on the FactKG dataset, reaches over **97%** on the WebQSP dataset, and is nearly **100%** on the Meta QA dataset.
>
> First and foremost, introducing the "Query Decoupling" step is essential. This step was designed to address challenges in semantic understanding and multi-hop reasoning for complex queries by decoupling intricate questions into manageable sub-queries. Attempting to retrieve and reason over a complete complex query in a single step often fails to produce accurate results, especially in the case of three-hop problems. While the decoupling process may introduce some errors, the proportion of such errors is controllable, and the benefits it brings in most cases significantly outweigh any potential drawbacks.
>
> Secondly, we apply strict quality control measures to the generated training data to minimize the possibility of introducing errors, as described in Section 3.4.1, "Quality Control." We have developed specific evaluation methods to ensure the quality of the generated data. Specifically, let $E$ represent the entity set of the original sentence, and $E_{div,i}$ denote the entity set for each sub-query after decoupling. The criteria for considering the decoupling results as high-quality are as follows:
>
> (a) $E_{div} \neq \emptyset$.
>
> (b) $E = \bigcup_{i=1}^{H} e_{div,i}$.
>
> (c) If $\( |E_{\text{div}}| > 1 \)$, then $\( \forall e_{div,i} \in E_{\text{div}}, e_{div,i} \subsetneqq E \)$. If $\( |E_{\text{div}}| = 1 \)$, then $\( E_{\text{div}} = E \)$.
>
> Overall, we aim for all entities in the original entity set to be reasonably and accurately assigned to each entity subset, ensuring that every query is appropriately decoupling.
>
>
>
>  **Comment 2:**
>
> Case analysis of scoring and filtering evidence triples using the expert model during the knowledge refinement stage.
>
> **Reply:**
>
> The expert model scores the evidence triples based on the query. In this step, conflicting knowledge with the query can be filtered out, while knowledge supporting the query is retained.
>
> For example, using data from FactKG:
>
> ```json
> {
>   "question": "Yes, Agra Airport is located in India where the leader is Narendra Modi.",
>   "types": [["coll:model", "num2", "multi claim"]],
>   "entity": ["India", "Agra_Airport", "Narendra_Modi"],
>   "Label": [true],
>   "triplet_evidence": [
>       ["Agra_Airport", "location", "India"],
>       ["India", "leader", "Narendra_Modi"],
>       ["Agra_Airport", "location", "Uttar_Pradesh"]
>   ]
> }
> ```
>
> The expert model scores the evidence triples, retaining those more relevant to the query, such as `(Agra_Airport, location, India)` and `(India, leader, Narendra_Modi)`, while discarding those that do not directly support answering the query, such as `(Agra_Airport, location, Uttar_Pradesh)`.
>
> ```json
> {
>   "qid": 5517,
>   "question": Abdul Taib Mahmud was born in the Kingdom of Sarawak and he was succeeded by Abdul Rahman Ya'kub.",
>   "types": [["written", "num2", "multi claim"]],
>   "entity": ["Abdul_Rahman_Ya'kub", "Kingdom_of_Sarawak", "Abdul_Taib_Mahmud"],
>   "Label": [true],
>   "used_all_relations": ["leader", "birthPlace", "placeOfBirth", "leader", "birthPlace", "placeOfBirth"],
>   "total_evidence": [
>       ["Abdul_Taib_Mahmud", "birthPlace", "Kingdom_of_Sarawak"],
>       ["Abdul_Taib_Mahmud", "successor", "Abdul_Rahman_Ya'kub"],
>       ["Abdul_Rahman_Ya'kub", "birthPlace", "Kingdom_of_Sarawak"],
>       ["Abdul_Taib_Mahmud", "children", "Sulaiman_Abdul_Rahman_Taib"],
>     ]
>   ]
> }
> ```
>
> The expert model scores the triples and selects `(Abdul_Taib_Mahmud, birthPlace, Kingdom_of_Sarawak)` and `(Abdul_Taib_Mahmud, successor, Abdul_Rahman_Ya'kub)`, while irrelevant noisy triples are filtered out to prevent interference with the reasoning results.

---

> > ### Author Response · Authors · 2024-11-22
> > **Response to Reviewer 8UTd （2/2）**
> >
> > **Comment 3:**
> >
> > Quantitative analysis of the noise propagation process and methods for noise control.
> >
> > **Reply:**
> >
> > We randomly selected 100 samples from the FactKG dataset and conducted a detailed analysis of noise introduction and reduction across the steps of decoupling, retrieval, scoring, and reconstruction.
> >
> > - **Noise introduction**: Refers to the introduction of incorrect knowledge, conflicting knowledge, or loss of correct information at a particular step.
> > - **Noise reduction**: Refers to successfully removing incorrect or irrelevant knowledge at a particular step.
> > - **Correctness**: Indicates whether the current knowledge information contains correct knowledge.
> >
> > |                    | Query Decoupling | Subgraph Retrieval | Knowledge Refinement | Knowledge Reconstruction |
> > | :----------------: | :--------------: | :----------------: | :------------------: | :----------------------: |
> > | Noise introduction |        9         |         24         |          2           |            3             |
> > |  Noise reduction   |        -         |         -          |          16          |            6             |
> > |    Correctness     |        92        |         87         |          85          |            84            |
> >
> > 1. In the **Query Decoupling** stage, certain cases may experience partial loss of entity information.
> > 2. In the **Subgraph Retrieval** stage, as we aim to retrieve as much knowledge relevant to the query as possible, it is inevitable to introduce some irrelevant knowledge and even knowledge that conflicts with correct information. Among them, some conflicting information may interfere with the results, while some irrelevant information has a minor impact.
> >
> > 3. In the **Knowledge Refinement** stage, some incorrect and irrelevant triples are scored and removed during the process, but a few correct answers may also be mistakenly filtered out.
> >
> > 4. In the **Knowledge Reconstruction** stage, while converting triples into textual information, the model performs implicit reasoning. During this process, the model may actively discard some incorrect or conflicting information and even correct erroneous information, but this may also result in the loss of a few correct pieces of information.
> >
> > When addressing complex problems, the introduction of noise is often unavoidable. Through the collaborative operation of various tasks, particularly during the **Knowledge Refinement** and **Knowledge Reconstruction** stages, we effectively control noise, significantly mitigating its cumulative effects across tasks and reducing its impact on overall performance. This further validates the robustness and effectiveness of our approach in complex knowledge reasoning scenarios.
> >
> >
> >
> > **Comment 4:**
> >
> > Discussion on generalization across different domains and applicability in diverse knowledge areas.
> >
> > **Reply:**
> >
> > We conducted experiments on both general-purpose and domain-specific datasets. FactKG and WebQSP utilize DBpedia and Wikidata, two large-scale knowledge graphs with extremely broad knowledge coverage, demonstrating RefKG's adaptability to complex question-answering tasks across multiple domains. Meanwhile, MetaQA, based on the MovieQA knowledge graph in the movie knowledge domain, further validates RefKG's exceptional performance in domain-specific tasks.
> >
> > We have summarized the knowledge graphs used in the three benchmark datasets, the corresponding knowledge domains, the number of triples required per query, the hop numbers of the queries, and the best performance of our method on these datasets, as detailed in the table below:
> >
> > | Benchmark |         Knowledge graph          | Domain  | Average number of triplets per query |  Hop num   | Accuracy |
> > | :-------: | :------------------------------: | :-----: | :----------------------------------: | :--------: | :------: |
> > |  FactKG   |  DBpedia (850 million triplets)  | General |                10.11                 |  1, 2, 3   |  84.04   |
> > |  WebQSP   | Wikidata (1.57 billion triplets) | General |                19.76                 | 1, 2, 3, 4 |   85.2   |
> > |  MetaQA   |     MovieQA（75k entities）      |  Movie  |                  2                   |  1，2，3   |   98.8   |
> >
> > It is worth emphasizing that our method incorporates knowledge-driven multi-task training, focusing on enhancing the model's ability to retrieve, refine, and apply knowledge, rather than limiting it to a specific knowledge domain. The capabilities learned by the model not only demonstrate broad generalization across various general-purpose domains but also support the plug-and-play integration of domain-specific knowledge graphs, enabling efficient performance on specialized tasks.
> >
> >
> >
> > We have revised the manuscript according to the Reviewer’s suggestion and response to each comment provided in the Weakness section above. We hope that our rebuttal aligns with the reviewer’s expectations, and we hope that the Reviewer can consider possibly giving a higher rating. Thanks.

---

### Official Review · Reviewer_We6j · 2024-11-04

**Soundness:** 3
**Presentation:** 3
**Contribution:** 2
**Rating:** 6
**Confidence:** 5

**Summary:**

This paper focuses on using large language models (LLMs) for knowledge graph question answering and fact verification tasks. The authors propose a framework that leverages an LLM to extract relevant reasoning paths from a knowledge graph and generate context based on these paths to reach the final answer. The framework involves three main steps: query decoupling, retrieval/construction/re-ranking of knowledge paths, and finally, context generation and question answering. They utilize GPT-3.5-turbo to generate training data for each of these steps and fine-tune smaller LLMs through multi-task learning.

**Strengths:**

- Overall, the proposed approach is straightforward and intuitive. Using an LLM to iteratively explore and retrieve reasoning paths from a knowledge graph is both novel and interesting.
- The method demonstrates strong empirical performance on two benchmark datasets, outperforming the baseline methods.
- A generated training dataset is provided, which could have potential value for future model training and evaluation.

**Weaknesses:**

- The proposed method’s novelty may be limited. The approach of using LLMs to decompose knowledge-intensive questions and then iteratively retrieve relevant information for knowledge-based tasks has already been widely discussed in existing literature, such as in the "self-ask" framework and its subsequent works. Additionally, using closed-source models like GPT-3.5-turbo to generate data is also common practice.
- I believe the computational cost of this method is a concern. To answer a multi-hop question, the model requires multiple LLM calls, often at least four. This cost may pose scalability challenges.
- The baseline models used have limitations. To better demonstrate the effectiveness of the additional steps in the proposed approach, a useful comparison would be a simple baseline that trains or prompts an LLM to generate possible reasoning paths from the knowledge graph, retrieves relevant paths, and uses them as context to answer questions. This would provide a clearer comparison of the value added by the additional steps in the proposed method.

[Self-Ask]: Measuring and Narrowing the Compositionality Gap in Language Models

**Questions:**

Is there an average number of LLM calls required to answer each question or verify each fact?

---

> ### Author Response · Authors · 2024-11-22
> **Response to Reviewer We6j (1/2)**
>
> Thank you for your valuable comments. We will explain your concerns point by point.
>
> **Comment 1:**
>
> The proposed method’s novelty may be limited.
>
> **Reply**:
>
> One of our major contributions is the design of a knowledge-driven multi-task instruction tuning method, which enables the model to effectively complete the full process of decoupling, exploration, refinement, reconstruction, and reasoning within a knowledge graph through multi-task collaboration. Multi-task fine-tuning allows the model to share learned features and representations across different tasks. This not only improves training efficiency and generalization capabilities but also enables the model to transfer knowledge gained from one task to others.
>
> Unlike traditional LLMs that primarily perform shallow understanding and generation of input content, our method endows the model with reflection capabilities based on knowledge graphs. Specifically, after training, the model can deeply evaluate the plausibility of information, actively identify and correct potential errors in knowledge. This capability surpasses traditional generative paradigms, enabling the model to perform deep reasoning in complex knowledge scenarios.
>
> Another significant contribution is that we trained an expert scoring model based on LLM. This module can identify and filter out triples that do not support question answering, effectively controlling the introduction of noise. This significantly enhances the accuracy of reasoning tasks across various knowledge scenarios. Additionally, this module improves the interpretability and efficiency of knowledge-based question-answering tasks.
>
>
>
> **Comment 2:**
>
> The computational cost poses scalability challenges, as well as the average number of LLM calls.
>
> **Reply:**
>
> Firstly, our method RefKG employs a process of decoupling, retrieval, refinement, and reasoning to enable LLMs to engage in deep thought on knowledge graphs. By invoking the LLM multiple times to accomplish various tasks, this approach is not redundant but essential for fully tapping into the LLM's potential for deep understanding and utilization of knowledge, ensuring both accuracy of results and effective use of knowledge in a way that is irreplaceable.
>
> KAPING[1] and KB-BINDER[2] make only a few calls to the large language model (LLM), including just one instance. We conducted a comparison with KAPING on WebQSP (wikidata) as shown in the following graph:
>
> |     Method      |        Model Size         | Number of calls | Accuracy |
> | :-------------: | :-----------------------: | :-------------: | :------: |
> |     KAPING      |           6.7B            |       few       |  53.34   |
> |     KAPING      |           175B            |       few       |  69.58   |
> |    KB-BINDER    | code-davinci-002(unknown) |       few       |   74.4   |
> | **RefKG(ours)** |            7B             |    multiple     | **85.2** |
>
> The results indicate that the approach of making only a few calls to the LLM fails to fully exploit the potential of the LLM to solve complex problems, thus not achieving optimal performance.
>
> Secondly, from a scalability perspective, RefKG employs knowledge-driven multitask instruction fine-tuning on LLMs, allowing a single LLM to exhibit multiple capabilities. With a one-time training and deployment, it can flexibly handle calls for various tasks. This approach not only conserves resources but also maintains the method's transferability and scalability.
>
> Finally, we conducted a detailed quantification of the scale and difficulty of different tasks, as well as the average number of times RefKG invoked LLMs and the inference speed. We randomly selected 100 samples from each dataset for experimentation, and the results are shown in the table below:
>
> | Benchmark | Average triplets numbers | Average number of calls | Average inference time |
> | :-------: | :----------------------: | :---------------------: | :--------------------: |
> |  FactKG   |          10.11           |           4.8           |          2.4s          |
> |  WebQSP   |          19.76           |           4.4           |          2.1s          |
> |  MetaQA   |            -             |           5.1           |          1.9s          |
> |  Average  |            -             |           4.8           |          2.1s          |
>
> Across the three datasets, the average number of LLM invocations was 4.8, and the average total inference time was 2.1 seconds.
>
> [1] Knowledge-Augmented Language Model Prompting for Zero-Shot Knowledge Graph Question Answering
>
> [2] Few-shot In-context Learning for Knowledge Base Question Answering

---

> > ### Author Response · Authors · 2024-11-22
> > **Response to Reviewer We6j (2/2)**
> >
> > **Comment 3:**
> >
> > Incorporate a simple baseline model for comparison and analyze the effectiveness of the additional steps.
> >
> > **Reply:**
> >
> > As shown in Table 5, we present the results of ablation experiments using Llama-2 on the FactKG dataset. By gradually removing individual tasks and observing the performance changes, we identified the following patterns:
> >
> > 1. **Removing the Knowledge Reconstruction task**: Directly reasoning with triples led to a **20.11%** performance decrease.
> > 2. **Retaining the Knowledge Reconstruction task without training it**: Performance decreased by **12.27%**.
> > 3. **Retaining the Knowledge Refinement task without training it**: Performance decreased by **2.71%**.
> > 4. **Retaining the Joint Inference task without training it**: Performance decreased by **30.64%**.
> >
> > These results clearly demonstrate the significant contribution of each task to the overall performance improvement.
> >
> > In addition, we conducted new comparative experiments using an untrained model to complete the entire process. In the experiments, **Base Model** represents the results obtained by directly using the untrained model, while **RefKG** represents the results achieved by applying our method. The experimental results are as follows:
> >
> > |   Model    | Base Model | RefKG(ours) | Difference |
> > | :--------: | :--------: | :---------: | :--------: |
> > |  Llama-2   |   34.12    |    81.26    |   -47.14   |
> > |   Bloom    |   37.65    |    84.04    |   -46.39   |
> > | Interlm-2  |   39.41    |    82.04    |   -42.63   |
> > | Baichuan-2 |   31.73    |    80.30    |   -48.57   |
> > |  Average   | **35.73**  |  **81.84**  | **-46.11** |
> >
> > The experimental results show that directly using the untrained **Base Model** leads to an average performance drop of **46.11%**. This indicates that untrained models struggle to handle our designed multi-task framework and are limited in their ability to tackle tasks involving complex knowledge.
> >
> > In contrast, **RefKG** significantly enhances the model's adaptability and performance through carefully designed tasks and targeted training. Our task design emphasizes knowledge reconstruction, refinement, and joint inference, with these steps working collaboratively to form a comprehensive reasoning mechanism. This enables the model to better handle complex knowledge scenarios and question-answering tasks.
> >
> >
> >
> > We have revised the manuscript according to the Reviewer’s suggestion and response to each comment provided in the Weakness section above. We hope that our rebuttal aligns with the reviewer’s expectations, and we hope that the Reviewer can consider possibly giving a higher rating. Thanks.

---

> > > ### Comment · Reviewer_We6j · 2024-11-27
> > >
> > > Thank you for your response. It addressed some of my concerns, and I will improve my score. Please consider adding this discussion to the final version.

---

### Author Response · Authors · 2024-11-22
**General Response**

We appreciate you taking the time to review our comments. We have received feedback from four reviewers, all of whom have provided thoughtful insights. Almost all the reviewers agree that our paper is well-structured, well-experimented, and easy to understand. However, the reviewers still had some concerns, and we have summarized these into four points, each of which we have analyzed and discussed in detail.



1. **Motivation and Contribution**:

Current approaches often introduce additional noise in the pipeline process of knowledge retrieval and reasoning, leading to the accumulation of errors, impeding LLMs from effectively combining the external knowledge in answering complex multi-hop questions. To this end, we introduce RefKG, an innovative framework specifically crafted to enhance the reasoning capabilities of LLMs through reflective engagement with knowledge graphs.

One of our major contributions is the design of a knowledge-driven multi-task instruction fine-tuning method, which enables the model to effectively complete the full process of decoupling, exploration, refinement, reconstruction, and reasoning within a knowledge graph through multi-task collaboration. Multi-task fine-tuning allows the model to share learned features and representations across different tasks. This not only improves training efficiency and generalization capabilities but also enables the model to transfer knowledge gained from one task to others.

Another significant contribution is that we trained an expert scoring model based on LLM. This module can identify and filter out triples that do not support question answering, effectively controlling the introduction of noise. This significantly enhances the accuracy of reasoning tasks across various knowledge scenarios. Additionally, this module improves the interpretability and efficiency of knowledge-based question-answering tasks.



2. **Quantitative analysis of the noise propagation process**

We randomly selected 100 samples from the FactKG dataset and conducted a detailed analysis of noise introduction and reduction across the steps of decoupling, retrieval, scoring, and reconstruction.

- **Noise introduction**: Refers to the introduction of incorrect knowledge, conflicting knowledge, or loss of correct information at a particular step.
- **Noise reduction**: Refers to successfully removing incorrect or irrelevant knowledge at a particular step.
- **Correctness**: Indicates whether the current knowledge information contains correct knowledge.

|                    | Query Decoupling | Subgraph Retrieval | Knowledge Refinement | Knowledge Reconstruction |
| :----------------: | :--------------: | :----------------: | :------------------: | :----------------------: |
| Noise introduction |        9         |         24         |          2           |            3             |
|  Noise reduction   |        -         |         -          |          16          |            6             |
|    Correctness     |        92        |         87         |          85          |            84            |

When addressing complex problems, the introduction of noise is often unavoidable. Through the collaborative operation of various tasks, particularly during the **Knowledge Refinement** and **Knowledge Reconstruction** stages, we effectively control noise, significantly mitigating its cumulative effects across tasks and reducing its impact on overall performance. This further validates the robustness and effectiveness of our approach in complex knowledge reasoning scenarios.



3. **More baseline comparison experiments**

We conducted new comparative experiments using an untrained model to complete the entire process. In the experiments, **Base Model** represents the results obtained by directly using the untrained model, while **RefKG** represents the results achieved by applying our method. The experimental results are as follows:

|   Model    | Base Model | RefKG(ours) | Difference |
| :--------: | :--------: | :---------: | :--------: |
|  Llama-2   |   34.12    |    81.26    |   -47.14   |
|   Bloom    |   37.65    |    84.04    |   -46.39   |
| Interlm-2  |   39.41    |    82.04    |   -42.63   |
| Baichuan-2 |   31.73    |    80.30    |   -48.57   |
|  Average   | **35.73**  |  **81.84**  | **-46.11** |

The experimental results show that RefKG significantly enhances the model's adaptability and performance through carefully designed tasks and targeted training. Our task design emphasizes knowledge reconstruction, refinement, and joint inference, with these steps working collaboratively to form a comprehensive reasoning mechanism. This enables the model to better handle complex knowledge scenarios and question-answering tasks.



Once again, we sincerely thank you for your involvement and thoughtful feedback!

---

### Meta-Review · Area_Chair_tQJ4 · 2024-12-20

**Metareview:**

This paper explores the application of large language models (LLMs) for knowledge graph question answering (KGQA) and fact verification. The authors propose a framework that integrates LLMs to extract reasoning paths from knowledge graphs and generate context for answering queries. The framework comprises three key steps: (1) query decoupling, (2) retrieval, construction, and re-ranking of knowledge paths, and (3) context generation and question answering. To enhance performance, the authors use GPT-3.5-turbo to generate training data for these steps and fine-tune smaller LLMs via multi-task learning.

However, most reviewers argue that this work lacks technical novelty, featuring a relatively simple pipeline. Its applicability beyond fact verification and KGQA tasks remains unexplored, and the benchmarks used are insufficient to demonstrate its broader utility. The framework may struggle to generalize in domains with sparse or specialized KGs, as its performance heavily depends on the completeness and accuracy of the underlying KGs, which can vary across domains. Additionally, the iterative retrieval and reflection process is computationally intensive, raising concerns about scalability for large-scale applications. Finally, the paper contains some typos, suggesting insufficient proofreading.

**Additional Comments On Reviewer Discussion:**

The reviewers engaged in a discussion with the authors, but most of the reviewers think that the work lacks innovation.

---

### Decision · Program_Chairs · 2025-01-22

Reject